# CroCoDiLight: Repurposing Cross-View Completion Encoders for Relighting

**Alistair J Foggin**[1] **& William A P Smith**[1,2]
[1]Department of Computer Science, University of York    [2]pxld.ai
{alistair.foggin,william.smith}@york.ac.uk

## Abstract

Cross-view completion (CroCo) has proven effective as pre-training for geometric downstream tasks such as stereo depth, optical flow, and point cloud prediction. In this paper we show that it also learns photometric understanding due to training pairs with differing illumination. We propose a method to disentangle CroCo latent representations into a single latent vector representing illumination and patch-wise latent vectors representing intrinsic properties of the scene. To do so, we use self-supervised cross-lighting and intrinsic consistency losses on a dataset two orders of magnitude smaller than that used to train CroCo. This comprises pixel-wise aligned, paired images under different illumination. We further show that the lighting latent can be used and manipulated for tasks such as interpolation between lighting conditions, shadow removal, and albedo estimation. This clearly demonstrates the feasibility of using cross-view completion as pre-training for photometric downstream tasks where training data is more limited. Project page: https://alistairfoggin.com/projects/crocodilight

## 1 Introduction

Cross-view completion (CroCo) (Weinzaepfel et al., 2022; 2023) has recently emerged as a promising pre-training proxy task for downstream problems in 3D geometric vision. The CroCo objective is to complete missing patches in an image given a second, overlapping view of the same scene, taken from a different viewpoint (see Fig. 1, left). In order to solve the task, the model must use cross attention to implicitly reason about correspondence, relative pose and depth in order to cross-project content from the complete to the masked image. CroCo is able to plausibly complete the missing patches while being trained on uncontrolled image pair collections with varying illuminations, suggesting that the model not only geometrically cross-projects but also *relights* the scene.

The hypothesis underlying our work (see Fig. 1, right) is that the CroCo encoder must implicitly estimate illumination and encode it in the patch embeddings. The cross-view decoder then removes lighting from the second view (i.e. *delights* the scene contents), geometrically cross-projects and then applies the lighting estimated from the observed patches in the masked image (i.e. *relights*). Crucially, CroCo's training objective is unique among self-supervised vision models in requiring implicit relighting capabilities. While other pretrained vision models learn rich visual representations, none are explicitly trained to handle the photometric transformations that CroCo encounters during its cross-view completion task. However, CroCo's patch embeddings contain this knowledge.

To test our hypothesis, we propose a model in which latent patch embeddings from the CroCo encoder are explicitly *disentangled* into 1. a single latent vector representing lighting and 2. lighting invariant per-patch latents representing static scene information (i.e. geometry and materials). Then a second model *recombines* a given lighting latent with local intrinsic latents back into the CroCo latent domain. We propose a method to train these two networks using pairs of images from the same view with different lighting and the same structure. In general, such fixed view/varying lighting images are harder to acquire than the uncontrolled pairs required for training CroCo. We demonstrate that this disentanglement can be learnt with datasets two orders of magnitude smaller than CroCo's original training, suggesting that the underlying photometric understanding is already present and requires only extraction rather than learning from scratch. Finally, we train a single view CroCo decoder to transform CroCo latent patches back into RGB space with high fidelity which can be done

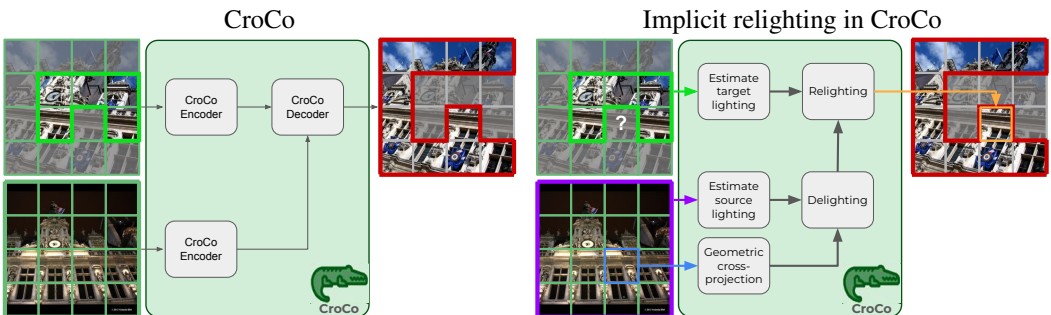

Figure 1: Left: data flow through CroCo. Right: the relighting that we hypothesise CroCo must implicitly perform when presented with a pair of images of the same scene but with different lighting. In order to predict a masked patch (shown with "?"), the target illumination must be estimated from the unmasked patches (green). Patches containing the same scene content (blue) must be delit using the source lighting estimated from the second view (purple) and relit (orange) using the estimated illumination. We show how to make this disentanglement explicit.

with any arbitrary image dataset. We name our method CroCoDiLight (Cross-view Completion for Disentangling Lighting).

Once trained, we demonstrate the use of our model for applications of lighting interpolation (temporal upsampling of timelapse videos) and relighting (timelapse illumination normalisation). We show that the learnt disentangling convincingly captures shading effects, cast shadows, coloured lighting, specularities and even local lighting. Going further, we show how to learn translations of the lighting latent to enable single image shadow removal and albedo estimation. We evaluate these translations on standard benchmarks and show performance competitive with the current state-of-the-art.

## 2 RELATED WORK

CroCo (Weinzaepfel et al., 2022; 2023) demonstrated the effectiveness of cross-view completion as a pre-training objective for purely *geometric* downstream tasks. The cross-attention decoder was fine-tuned for stereo depth and optical flow models. Subsequently, DUST3R (Wang et al., 2024) and MAST3R (Leroy et al., 2024) expanded upon this to directly predict a 3D point cloud using unposed frames. Our hypothesis is that the encoder from CroCo pre-training has also learnt lighting information from the scene, enabling *photometric* tasks such as relighting, shadow removal, and intrinsic image decomposition. Our work is similar in spirit to other efforts to repurpose foundation models for tasks where it is believed the original training objective means the model must already implicitly solve the task. For example, Kerssies et al. (2025) show that Vision Transformers implicitly learn segmentation, while generative image models have been shown to learn monocular depth estimation (Ke et al., 2024), complete intrinsic image decomposition (Ke et al., 2025) and zero-shot classification (Li et al., 2023a).

**Relighting** When it comes to relighting images, there are a few main approaches. HDR panoramas can be used to condition changes to the lighting of an image as done in UniRelight (He et al., 2025). Text conditioning is also possible in models such as Neural Gaffer (Jin et al., 2024) and IC-Light (Zhang et al., 2025). This is most effective in outdoor scenes or for relighting individual objects where the global lighting can vary according to the aforementioned inputs. Other methods based on generative models can better handle local image dynamics with indoor lighting. An example of this is LightLab (Magar et al., 2025) which enables detailed control of the colour and intensity of individual lights. LumiNet (Xing et al., 2025) predicts lighting from a reference image and encodes it to a latent vector which is used to relight another image. All of these approaches enable relighting, often for perceptually good generation of new lighting from conditioning, but our method aims at developing a general-purpose model to disentangle the real lighting of an image in order to carry out lighting interpolation and transfer, along with more clearly defined tasks.

**Intrinsic Image Decomposition**   Beyond directly relighting images, many methods such as Ordinal Shading (Careaga & Aksoy, 2023) and Lossless Intrinsic Image Decomposition (Sha et al., 2025) tackle the task of intrinsic image decomposition into albedo and shading images. These two images can be easily recombined to produce the original image. Other methods such as DiffusionRenderer (Liang et al., 2025) separate and recombine images into further components such as ambient occlusion, normals, and depth. However, the primary challenge for all of these methods is training data. Synthetic datasets such as CGIntrinsics (Li & Snavely, 2018b) and ML-Hypersim (Roberts et al., 2021) render out dense decompositions but still leave the gap to real-world images. The Multi-Illumination dataset (Murmann et al., 2019) is a collection of fixed-camera real-world scenes, each with 25 controlled lighting conditions. BigTime (Li & Snavely, 2018a) on the other hand is a collection of in-the-wild timelapses with varying illumination conditions. While these provide training data, they don't provide direct supervision. Intrinsic Images in the Wild (IIW) (Bell et al., 2014) and Shading Annotations in the Wild (SAW) (Kovacs et al., 2017) provide sparse annotations of lighting conditions for real-world images along with corresponding evaluation benchmarks.

**Shadow Removal**   The task of shadow removal is challenging as shadows vary in their sharpness and in the way that they are cast. They can be cast from occluders outside of the image, or self-occlusion from objects within the image. Datasets provide supervision for these various shadow types. These include SRD (Qu et al., 2017), WSRD (Vasluianu et al., 2023), and ISTD (Wang et al., 2018) which provide image pairs of shadowed and shadow-free images that can be used for direct supervision. With these datasets, there are issues in colour and pixel alignment. To help tackle this, WSRD+ (Vasluianu et al., 2024) and ISTD+ (Le & Samaras, 2019) are modified versions which help fix those issues. Many current methods use an input mask to remove a specific shadow as done in HomoFormer (Xiao et al., 2024) and ShadowFormer (Guo et al., 2023), whereas other methods such as OmniSR (Xu et al., 2025a) remove shadows from the image directly without a mask.

## 3   RELIGHTING IN CROCO LATENT SPACE

Our method begins by encoding images with the pretrained (and frozen) CroCo v2 encoder (Weinzaepfel et al., 2023). An input image $X \in \mathbb{R}^{H \times W \times 3}$ is decomposed into a set of $N$ non-overlapping patches, each of size $P \times P$, such that the patches cover the entire frame, i.e. $N = HW/P^2$. The RGB patches are flattened to vectors, $\mathbf{x}_p \in \mathbb{R}^K$ with $p = 1 \ldots N$ representing spatial position and $K = 3P^2$, and embedded to the dimensionality $D = 1024$ used throughout the transformer models. The CroCo v2 encoder operates on patches of size $P = 16$, applies RoPE positional embeddings (Su et al., 2024) to the patches and encodes with a transformer:

$$\mathbf{z} = \mathcal{E}(\mathbf{x}), \tag{1}$$

where $\mathbf{x} = \text{patchify}(X) = \{\mathbf{x}_1, \ldots, \mathbf{x}_N\}$ and $\mathbf{z} = \{\mathbf{z}_1, \ldots, \mathbf{z}_N\}$ with $\mathbf{z}_i \in \mathbb{R}^D$. The CroCo encoder was originally trained on images of resolution $H = W = 224$. However, since RoPE can generalise to patch coordinates beyond the positions seen during training, we operate at resolution $H = W = 448$ throughout our method, similar to how DUST3R (Wang et al., 2024) scaled up resolution during training and used a sliding window approach for scaling up further.

Our approach (see Fig. 2) starts with a *delighting transformer* which disentangles illumination from scene intrinsic properties by translating the patch latents into intrinsic patches and estimating a lighting latent vector that describes the appearance in that particular illumination environment. Second, a *relighting transformer* which recombines a lighting latent vector with intrinsic patches producing patch embeddings in the original CroCo latent space. Finally, to ensure high quality image synthesis we train a *single view decoder* to transform from CroCo latent space back to RGB images.

### 3.1   SINGLE VIEW DECODER

The original CroCo decoder was trained for the difficult task of using cross-attention to predict masked patches in an image from a second view. As the decoder is binocular, the only way to decode a single view is to remove masking and use the same encoded image as if it were the second view. It was also trained with supervision only on the reconstruction of masked patches, not the whole image. This means that the image quality is low when decoding a single view (Appendix B).

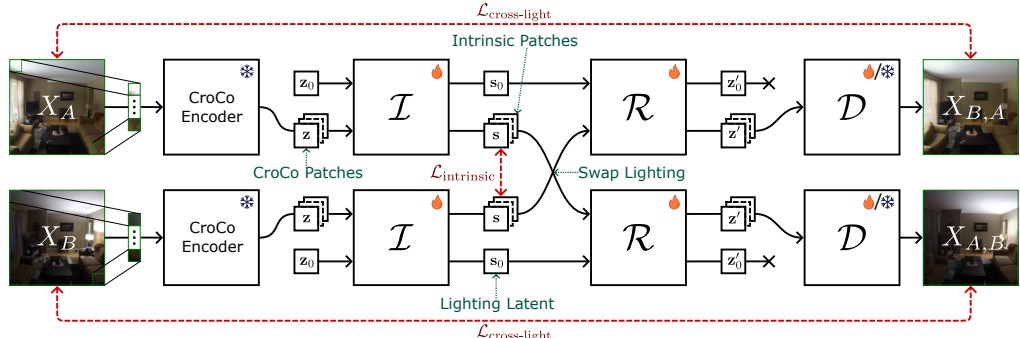

Figure 2: The architecture of the model comprises four main components. First is the frozen CroCo encoder. Last is the decoder $\mathcal{D}$ which is separately pre-trained and then frozen to decode from CroCo latent space to RGB. Then there are the delighting and relighting transformers, $\mathcal{I}$ and $\mathcal{R}$ respectively, which disentangle lighting and intrinsics before recombining them. The training process here shows pairs of images encoded and relit to match the lighting of the other image.

Our downstream applications of relighting, shadow removal and albedo estimation all operate on single images and require high quality image output. We therefore train our own single view decoder with an autoencoder objective: i.e. we train the decoder to reconstruct original images from embeddings given by the frozen CroCo encoder. Specifically, we train a decoder:

$$X' = \mathcal{D}(\mathbf{z}), \tag{2}$$

which uses self-attention with a DPT head (Ranftl et al., 2021) for reconstruction. The self-attention module consists of 12 layers each with 16 heads. Every layer is followed by a 2-layer MLP with a hidden layer size of $2D$. To ensure high-fidelity reconstruction, we train the decoder using a combination of perceptual (LPIPS (Zhang et al., 2018)) and mean squared error (MSE) losses:

$$\mathcal{L}_{\text{img}}(X, X') = \lambda \mathcal{L}_{\text{LPIPS}}(X, X') + (1 - \lambda)\mathcal{L}_{\text{MSE}}(X, X'), \tag{3}$$

where we use $\lambda = 0.5$ throughout. This objective does not require paired images so can be trained on any image dataset, for which we use ImageNet (Deng et al., 2009; Russakovsky et al., 2015).

### 3.2 DELIGHTING TRANSFORMER

In order to estimate illumination, we append a learnable query vector $\mathbf{z}_0 \in \mathbb{R}^D$ to the CroCo patch latent embeddings: $\hat{\mathbf{z}} = \{\mathbf{z}_0, \ldots, \mathbf{z}_N\}$. We pass this augmented set to the delighting transformer to disentangle lighting:

$$\hat{\mathbf{s}} = \mathcal{I}(\hat{\mathbf{z}}), \tag{4}$$

where $\hat{\mathbf{s}} = \{\mathbf{s}_0, \ldots, \mathbf{s}_N\}$. Here, $\mathbf{s}_0$ contains lighting information for the whole image while $\mathbf{s} = \{\mathbf{s}_1, \ldots, \mathbf{s}_N\}$ contains intrinsic information about the original image patches, i.e. with the effect of lighting removed. The architecture of $\mathcal{I}$ closely follows the architecture of the CroCo encoder. It comprises 8 consecutive self-attention blocks, each with 16 heads. Every self-attention block is followed by a 2-layer MLP with a hidden size double the dimensions of a single patch. Each of the patches retains the same RoPE positional encoding used by the CroCo encoder. As the self-attention blocks expect positional encoding for every input patch, when we append $\mathbf{z}_0$, we give it a unique positional encoding for the position $-1$ as no encoded patch will have that position.

### 3.3 RELIGHTING TRANSFORMER

Next, we train a relighting transformer that recombines a lighting latent vector with the intrinsic patch embeddings to return to the CroCo latent space:

$$\hat{\mathbf{z}}' = \mathcal{R}(\hat{\mathbf{s}}), \tag{5}$$

where $\hat{\mathbf{z}}' = \{\mathbf{z}_0', \ldots, \mathbf{z}_N'\}$. We discard $\mathbf{z}_0'$ and retain only the patch embeddings in CroCo latent space, $\mathbf{z}' = \{\mathbf{z}_1', \ldots, \mathbf{z}_N'\}$. Note that the input lighting latent vector, $\mathbf{s}_0$, need not be from the original image but could be extracted from another image for the purposes of relighting. The architecture of $\mathcal{R}$ comprises an identical architecture to that of the delighting transformer, $\mathcal{I}$, but only keeps $\mathbf{z}'$ as the predicted CroCo latent patch embeddings to be decoded.

### 3.4 PAIRED IMAGE TRAINING

We now explain how $\mathcal{I}$ and $\mathcal{R}$ can be trained using paired images. While CroCo used pairs of images of the same scene from different viewpoints (possibly with different lighting), we use pairs of pixel-aligned images from the same viewpoint where the illumination, and therefore appearance, is different. This provides two forms of supervision.

First, the two images $X_A$ and $X_B$ are encoded with the CroCo encoder and then disentangled by the delighting transformer into lighting latents, $\mathbf{s}_0^A$ and $\mathbf{s}_0^B$, and patch-wise intrinsic latent embeddings, $\mathbf{s}^A$ and $\mathbf{s}^B$. Since we expect the intrinsic latent embeddings to represent static aspects of the scene such as geometry and materials, we use an MSE loss to encourage consistency between the intrinsic patches from the two images:

$$\mathcal{L}_{\text{intrinsic}} = \frac{1}{N}\sum_{i=1}^{N}\left\|\mathbf{s}_i^A - \mathbf{s}_i^B\right\|^2. \tag{6}$$

Second, we can impose a cross-lighting constraint. Namely, we can relight the intrinsic patches from image A with the lighting estimate from image B:

$$X'_{A,B} = \mathcal{D}(\mathcal{R}(\{\mathbf{s}_0^B, \mathbf{s}_1^A, \ldots, \mathbf{s}_N^A\})), \tag{7}$$

and similarly using the lighting from B with the intrinsic patches from A to produce $X'_{B,A}$. This allows us to define a cross-lighting loss:

$$\mathcal{L}_{\text{cross-light}} = \mathcal{L}_{\text{img}}(X_A, X'_{B,A}) + \mathcal{L}_{\text{img}}(X_B, X'_{A,B}). \tag{8}$$

We train the delighting and relighting modules using a weighted sum of the above two losses. Together, they force the delighting transformer to bottleneck lighting-related aspects of appearance through the lighting latent vector while making the intrinsic patch embeddings invariant to illumination. Meanwhile, the relighting transformer is encouraged to combine lighting and intrinsic information such that decoded images closely match the originals.

### 3.5 SLIDING WINDOW FOR HIGH-RESOLUTION RELIGHTING

In order to generalise the outputs of the model to high-resolution images, each image must be split into multiple overlapping $448 \times 448$ tiles that are fed into the model as a batch, relit with a lighting latent per-tile, and merged together by blending the overlapping pixels. While the RoPE embeddings do enable resolution-agnostic self-attention, the lighting latent has been trained to compress the lighting information of the specified resolution. This means that despite the intrinsic latent vectors potentially being accurate for arbitrarily sized images, the lighting latent vector is only meaningful at the specified resolution. The sliding window ensures that the lighting latent is optimally used.

## 4 DATASET

Training the lighting modules requires pixel-aligned, paired images with illumination variation. In general, this type of data is challenging to collect, but as long as this requirement is met, many different datasets can be incorporated in training, whether or not illumination and shadows are controlled.

**Uncontrolled lighting**   For uncontrolled illumination and shading, BigTime (Li & Snavely, 2018a) is a set of in-the-wild timelapses from static cameras. It contains both indoor and outdoor scenes and helps models have a more general understanding of lighting. Despite its limited size, it provides valuable information and can be scaled up as static timelapses are often captured and put online. These can easily be incorporated into the training process. The main restriction in expanding this type of dataset is ensuring that the timelapses collected retain consistent structure, as often they are of the sky or of crowds of people, both of which have changing content beyond just lighting.

**Controlled lighting**   The Multi-Illumination dataset (Murmann et al., 2019) consists of around 1000 scenes each with a set of 25 controlled lighting conditions. All the scenes are indoor but contain a variety of challenging materials to model, including transparent and highly reflective objects. From these two datasets, we kept the Multi-Illumination test-set aside for evaluation in our experiments,

and used the rest in training. For a single epoch through these datasets, every single image was included once, and a random pair from the same scene was randomly selected every iteration. While this does mean that several images per scene may be seen multiple times in an epoch, it is still relighting it to a different illumination condition and it guarantees that every image is seen.

Beyond just arbitrary pairs of various lighting and exposures, albedo estimation and shadow removal datasets were beneficial. Three shadow removal datasets, SRD, ISTD+, and WSRD+ were included in training using pairs of shadowed and shadow-free images with which lighting was swapped. Synthetic image pairs of rendered and albedo images were included to encourage further extraction of shading information into the lighting latent, in particular ML-Hypersim and CGIntrinsics. The ML-Hypersim dataset includes series of images following a camera trajectory within a scene, but for the purposes of this paper and reducing overlapping images, only the first frame of each camera trajectory was included. Overall, these datasets form 57k image pairs (36k real and 21k synthetic) to be used in training, which is two orders of magnitude less than the required data for CroCo v2 (Weinzaepfel et al., 2023) which used 2M synthetic pairs and 5M real pairs.

**Augmentation** In order to help prevent over-fitting to the training images, data augmentation was used. The primary augmentation was random $448 \times 448$ crops of the original images. This meant that different sections of the image are seen every epoch, and that it has seen partial images which helps it generalise to various resolutions when using a sliding window. We also incorporated random conversion to greyscale to highlight differences between shadows and colour variation. In general, any augmentation that preserves the pixel-wise alignment is possible.

## 5 MANIPULATIONS IN THE LIGHTING LATENT SPACE

The cross-lighting training described above disentangles intrinsic patch latent vectors from a single dynamic lighting latent vector. This opens up the possibility of editing images by manipulation within the lighting latent space. Given the aligned nature of the paired training images, the lighting latent explains lighting effects in *image space* as opposed to world space (see Appendix C). To further explore how it works, several downstream tasks were explored.

### 5.1 LIGHTING INTERPOLATION AND RELIGHTING

The most obvious lighting manipulation is to relight images via interpolation between different lighting conditions. Given that the latent vector works in image space, the examples explored are

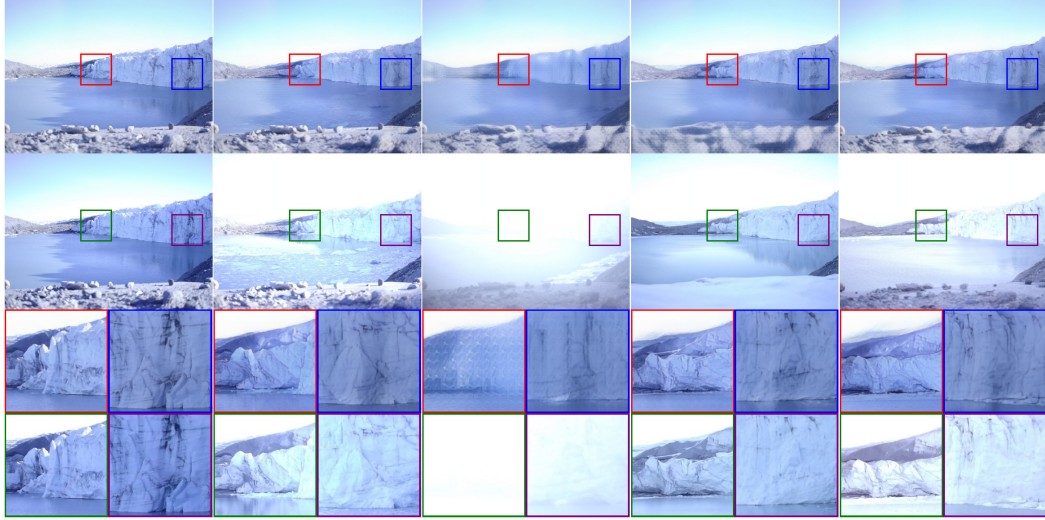

Figure 3: A series of time-lapse photos of a glacier with varying lighting conditions relit to match the lighting condition of the first image. The top row demonstrates transferring the lighting of the first image while still showing the intrinsic changes. The second row is of the original images, and the final row is of the magnified colour-coded patches.

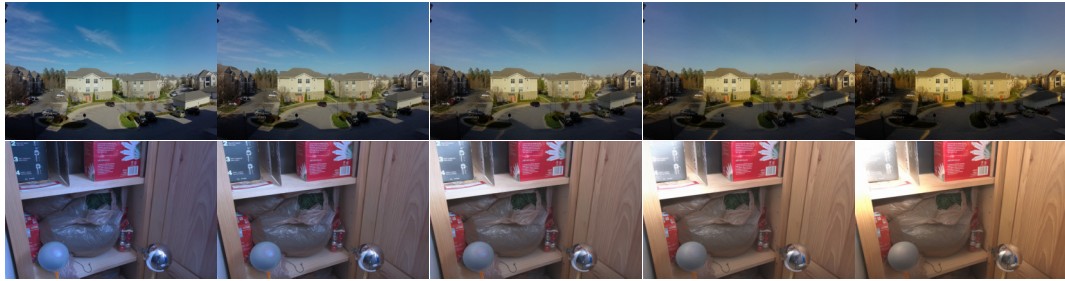

Figure 4: A series of images demonstrating capabilities of the latent-space in changing shadows and specular highlights. This is done with linear interpolation between two input lighting conditions.

using fixed camera positions with pixel-wise alignment. The main use of this is manipulating the lighting in timelapses and other fixed-view sets of images with various lighting conditions.

**Lighting stabilisation in timelapse**    One challenge with timelapses is that the lighting can change drastically from photo to photo, meaning it can be more challenging to see what intrinsic components of the underlying scene are changing. To tackle this, we can take a timelapse which has desirable lighting conditions (e.g. well exposed, ambient as opposed to saturated) at the start, extract the lighting latent, and apply that lighting latent to all subsequent images in the timelapse as shown in Figure 3 (and see supplementary video). This provides consistent lighting throughout, with only the structure contained in the intrinsic latent patches changing.

**Temporal timelapse upsampling**    Other components of the lighting latent vector are shadows and specularities. To better understand these components, the lighting latent vectors were extracted from various keyframes that have distinct lighting conditions. We then linearly interpolated between these latent vectors to compare with direct interpolation in RGB space. Figure 4 demonstrates the extent to which shadows, specularities, and lighting are embedded in the latent space. While the motion of the shadows is not entirely smooth, it demonstrates that all the correct lighting conditions exist within that space and the trajectory can be learnt. We provide additional interpolation and relighting results in Appendix D. This lighting interpolation enables temporal upsampling to generate intermediate frames. This allows for extending the length of such timelapses in post-production. We show some quantitative results using FloLPIPS (Danier et al., 2022) in Table 1 in which we interpolate halfway between every 7th frame and evaluate the temporal perceptual quality.

|                       | Clock Shadows | Day-night Cycle | Indoor Shadows |
| :---:                 | :---:         | :---:           | :---:          |
| Interpolation Method  | FloLPIPS ↓    | FloLPIPS ↓      | FloLPIPS ↓     |
| Image Space           | 0.309         | **0.041**       | 0.950          |
| Latent Space (Ours)   | **0.286**     | 0.043           | **0.923**      |

Table 1: A quantitative evaluation of temporal timelapse upsampling. Every 7 frames were sampled, and interpolated to halfway between the two, either in image space, or in lighting latent space. The triplet of ground truth frames on either side, and the interpolated frame compared to the reference frame was evaluated using FloLPIPS. These results were averaged across each timelapse. As we are testing lighting latent-space interpolation, intrinsics are fixed for each triplet, which means that any structural dynamics are detrimental as image-space interpolation can smooth those dynamics.

## 5.2   LEARNING LIGHTING LATENT TRANSFORMATIONS

Beyond lighting interpolation, temporal upsampling, and lighting stabilisation, more concrete downstream tasks also exist to which our model can be applied. As we have seen, lighting and shadows can be manipulated using latent vector interpolation. Transformations in latent space exist for shadow removal and albedo reflectance estimation which models can be trained to learn.

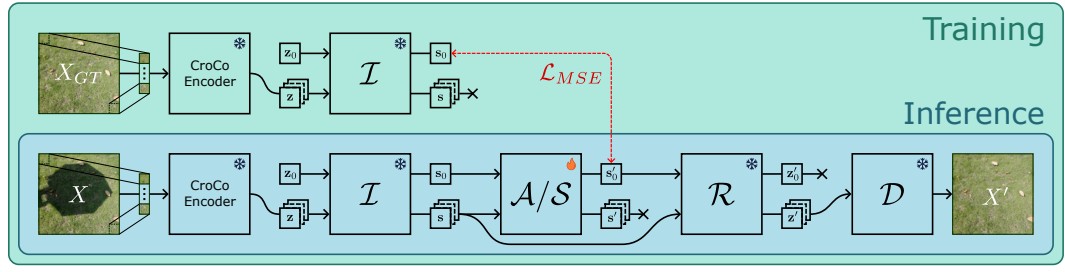

Figure 5: Diagram describing the training process for both the shadow removal and albedo estimation components $\mathcal{S}$ and $\mathcal{A}$ to learn the transformations in the lighting latent space. We distinguish between which steps are only done during training and which are done in training and inference. We also show that all the components apart from $\mathcal{S}/\mathcal{A}$ have frozen weights during training.

**Shadow Removal** For the task of shadow removal, a model $\mathcal{S}$ matching the architecture of $\mathcal{R}$ and initialised by $\mathcal{R}$'s weights was trained to map from a shadowed latent $\mathbf{s}_0$ to a shadow-free latent $\mathbf{s}_0'$ while using the intrinsic latent patches $\mathbf{s}$ to guide the process. The difference to $\mathcal{R}$ in processing $\hat{\mathbf{s}}$ is that we keep the output $\mathbf{z}_0'$ as $\mathbf{s}_0'$ and discard $\mathbf{z}'$ now called $\mathbf{s}'$ (see $\mathcal{S}$ in Figure 5). The output latent is then appended to the original intrinsic latents to produce $\hat{\mathbf{s}}' = \{\mathbf{s}_0', \mathbf{s}_1, \ldots, \mathbf{s}_N\}$ which can be subsequently re-entangled using $\mathcal{R}$ and decoded. The loss function is mean squared error (MSE) in the lighting latent space to match the encoded lighting latent from the ground truth shadow-free image. At this point, only $\mathcal{S}$ was being trained. All the other components had frozen weights. The training process is shown in Figure 5. This was supervised on the training splits of the SRD, ISTD+, and WSRD+ datasets. We also compared the model to using an optimal shadow-free latent which we extract from the shadow-free image and apply it to the intrinsic latents of the shadowed latent. While in practice, this does not work as the shadow-free image is not available, it demonstrates that the mapping exists within the latent space and our pre-training task is capable produce competitive results compared to models trained specifically on shadow removal.

For evaluating our method, we apply the sliding window, removing the shadow from the each tile. After recombining the tiles, we resize the output to $256 \times 256$ as done in previous methods before calculating the metrics. The metrics we report are LPIPS (Zhang et al., 2018), Mean Absolute Error (MAE) in L*a*b* colour-space, structural similarity (SSIM), and Peak Signal-to-Noise Ratio (PSNR) as shown in Table 2. There are several inconsistencies in previous methods in whether they were using Root Mean Squared Error (RMSE) or MAE, making results less reliable than SSIM and PSNR. We therefore re-evaluated all the methods that provided their output shadow-free images. Where these images were not available, we show the metrics reported in the paper.

Albedo Estimation        Shadow Removal

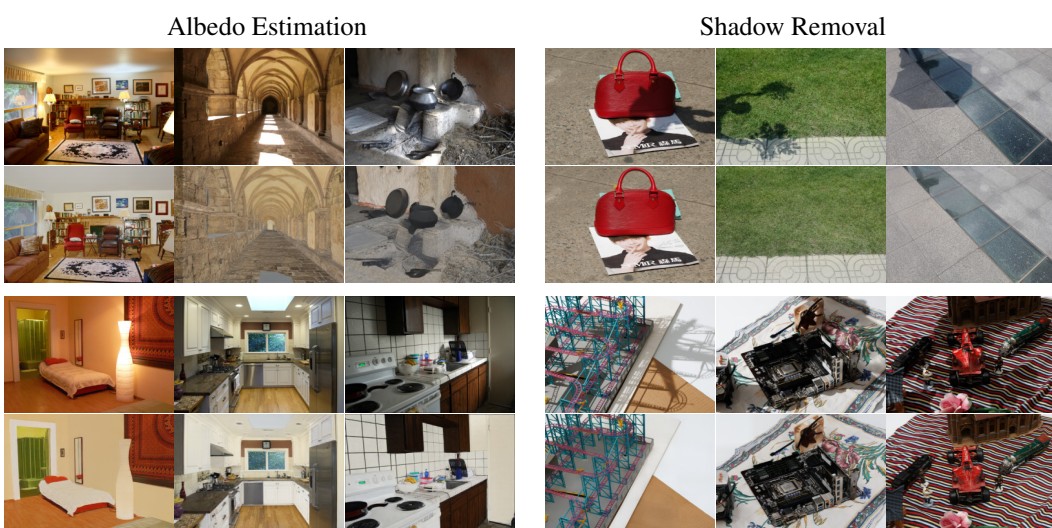

Figure 6: Examples of our albedo estimation and shadow removal.

| Method | Mask | ISTD+ | | | | SRD | | | |
|---|---|---|---|---|---|---|---|---|---|
| | | LPIPS ↓ | MAE ↓ | SSIM ↑ | PSNR ↑ | LPIPS ↓ | MAE ↓ | SSIM ↑ | PSNR ↑ |
| BMNet | Yes | 0.027 | 1.78 | 0.967 | 33.92 | 0.047 | 2.36 | 0.944 | 31.92 |
| ShadowFormer | Yes | 0.022 | 1.60 | 0.971 | 35.37 | - | - | 0.958* | 32.90* |
| Li et al. | Yes | 0.033 | 2.01 | 0.959 | 33.72 | 0.044 | 2.17 | 0.942 | 33.72 |
| HomoFormer | Yes | 0.022 | 1.56 | 0.968 | 35.26 | 0.035 | 1.56 | 0.955 | 35.33 |
| DMTN | No | 0.033 | 2.16 | 0.960 | 32.19 | 0.045 | 2.29 | 0.937 | 32.79 |
| ShadowRefiner | No | 0.043* | - | 0.928* | 31.03* | - | - | - | - |
| OmniSR | No | 0.025 | 1.83 | 0.966 | 33.30 | 0.042 | 2.41 | 0.941 | 31.96 |
| StableShadowRemoval | No | 0.021 | 1.67 | 0.968 | 35.10 | 0.033 | 2.21 | 0.944 | 33.24 |
| CroCoDiLight (ours - $\mathcal{S}$) | No | 0.038 | 2.86 | 0.929 | 30.17 | 0.041 | 3.01 | 0.931 | 30.01 |
| CroCoDiLight (ours - oracle) | No | 0.028 | 2.00 | 0.936 | 33.41 | 0.034 | 2.25 | 0.937 | 32.47 |

Table 2: Shadow Removal evaluations on both the ISTD+ and SRD datasets. We distinguish between mask-based models and those that do not require shadow masks. All metrics are re-evaluated on the provided images from each model unless indicated otherwise. "*" indicates that the numbers are self reported and cannot be re-evaluated. "-" indicates that they did not report the metric.

The methods we compare against are split into two categories, masked and unmasked. The majority of methods for shadow removal make use of an input mask to specify a specific shadow to remove. Recent methods have also been tackling the more challenging mask-free shadow removal task. While the metrics do not put our method as state of the art, our results in Figure 16 show examples of more effective shadow removal than other methods. It is primarily subtle colour variation across each image that is detrimental to the metrics. Our pre-training disentangles shadows into the lighting latent which is then transformed to a shadow-free latent. The oracle metrics demonstrate that better transformations do also exist. It is also worth noting that other methods fine-tune their models on case-by-case basis for each benchmark. This enables them to achieve better evaluation metrics, but limits the generalisability of their trained models. These differences are due to inherent visual biases in each dataset. Our method is jointly trained on all the aforementioned datasets and applied to every benchmark to produce a single general-purpose shadow-removal component $\mathcal{S}$. The masked methods we evaluate are BMNet (Zhu et al., 2022), ShadowFormer (Guo et al., 2023), the method from Li et al. (2023b), and HomoFormer (Xiao et al., 2024). The mask-free methods we evaluate are DMTN (Liu et al., 2023), ShadowRefiner (Dong et al., 2024), OmniSR (Xu et al., 2025a), and StableShadowRemoval (Xu et al., 2025b).

**Intrinsic Image Decomposition**  We also trained a latent-space transformation model $\mathcal{A}$ for intrinsic image decomposition in predicting albedo reflectance images from a fully lit image. As with shadow removal, $\mathcal{A}$ retains the same architecture as $\mathcal{R}$ and is initialised with the same weights. It maps the lighting latent vector $s_0$, as part of $\hat{s}$, to the new albedo latent $s_0'$ to be re-entangled with the intrinsic latents $s$ using $\mathcal{R}$ as shown in Figure 5. For this task, we train the model using the CGIntrinsics and ML-Hypersim datasets. Note that while this was only trained using synthetic data, the real data in pre-training enables generalisability.

In comparing against other models, we evaluate on the IIW WHDR benchmark using the test set defined by Narihira et al. (2015). In Table 3 we compare against CGIntrinsics (Li & Snavely, 2018b), NIID-Net (Luo et al., 2020), PIE-Net (Das et al., 2022), Ordinal Shading (Careaga & Aksoy, 2023), and IntrinsicDiffusion (Luo et al., 2024). Our model provides state-of-the-art results despite albedo estimation not being the primary task. We do not show methods trained on IIW for a fair evaluation as they often over-fit to the WHDR metric, our method is still very competitive against them despite not training on IIW. GLoSH (Zhou et al., 2019) has a score of 15.2%, and Lossless Intrinsic Image Decomposition (Sha et al., 2025) scores 13.8%. However, this evaluation metric does have some flaws which are

| Method | WHDR (%) ↓ |
|---|---|
| CGIntrinsics (2018) | 17.8 |
| NIID-Net (2020) | 16.6 |
| PIE-Net (2022) | 21.3 |
| Ordinal Shading (2023) | 24.9 |
| IntrinsicDiffusion (2024) | 17.9 |
| CroCoDiLight (Ours) | **15.4** |
| Ordinal Shading + 0.5 | 15.3 |
| CroCoDiLight (Ours) + 0.5 | **14.3** |

Table 3: Evaluations of the WHDR metric on the IIW test set for models that have not been trained on the IIW dataset.

highlighted in Ordinal Shading. They demonstrate that despite having qualitatively good results, they perform poorly on the benchmark, but by doing a simple shift by adding 0.5 to the image RGB values shifting it out of the range of 0 to 1, they get a significant jump in performance. This same arithmetic shift gives us a smaller jump in performance. Despite this potential for tuning a model to the benchmark, our method performs well without doing so. Figure 6 demonstrates our decomposition results beyond just metrics.

## 6    DISCUSSION

**Ablation Study**   In order to determine the effectiveness of CroCo pre-training, we trained two comparison models with the same data, number of iterations, and proposed model components $\mathcal{I}$ and $\mathcal{R}$. One was trained using the same architecture, with the CroCo encoder and our decoder, both jointly trained from scratch with $\mathcal{I}$ and $\mathcal{R}$. The other was without the CroCo encoder to simplify what the model had to learn by reducing the number of parameters. We instead added a simple encoder which splits the image into patches $\mathbf{x}$ and embeds them with a simple linear and normalisation layer into $\mathbf{z}$. Then to decode, a DPT head was attached to the relighting transformer $\mathcal{R}$ instead of using $\mathcal{D}$. The shadow removal and albedo estimation latent-space models $\mathcal{S}$ and $\mathcal{A}$ were also retrained for both comparisons. Table 4 demonstrates that when training everything from scratch, the simpler architecture does well. However, using the pre-trained CroCo v2 encoder makes notable improvements in albedo estimation and as well as improvements in shadow removal. The improvements on these metrics show the effectiveness of cross-view completion as pre-training for relighting tasks, primarily in embedding scene intrinsics in latent space. This provides significant evidence for our hypothesis of the photometric and relighting capabilities of CroCo latent space.

| CroCo | Pre-trained | IIW | | ISTD+ | | SRD | |
|---|---|---|---|---|---|---|---|
| | | WHDR ↓ | WHDR (+0.5) ↓ | SSIM ↑ | PSNR ↑ | SSIM ↑ | PSNR ↑ |
| | | 19.1% | 16.8% | 0.924 | 29.71 | 0.928 | 29.87 |
| ✓ | | 27.7% | 22.6% | 0.795 | 25.34 | 0.824 | 22.98 |
| ✓ | ✓ | **15.4%** | **14.3%** | **0.929** | **30.17** | **0.931** | **30.01** |

Table 4: Ablation study on the effectiveness of using the CroCo v2 encoder, with or without the pre-trained weights, or using a simpler architecture trained from scratch.

**Limitations and Future Work**   There are some limitations in our work, mostly deriving from the architectural decision to use a single lighting latent for each tile. The compression of the small latent vector means that reconstructing sharp shadows is difficult. Some bottleneck is necessary for disentangling lighting from intrinsics, but it inhibits the fidelity of lighting extraction. Tiling high-resolution images produces colour variations in shadow removal, as shown in Appendix F.1, which is due to a lack of global image context per-tile. Pixel-aligned training also means that lighting is extracted in image-space without 3D reasoning, leading to sensitivity in misalignment, and a current lack of control of relighting.

Architectural ablation studies to explore various lighting representations and alternative encoders would help to expand upon this work. Such lighting representations could consist of using multiple lighting latents, varying their size, or using spatially varying lighting maps. Comparisons of the CroCo v2 encoder with other pretrained encoders and their pretraining objectives would highlight differences in learning lighting invariance as opposed to understanding lighting changes. With exploring our model's capabilities, a lighting latent mapper could be trained to better guide interpolation when trained on videos. There is also the expansion to be done into user-controllable lighting from arbitrary images or other conditioning.

**Conclusion**   All these various ways of manipulating the lighting latent space for the purposes of relighting, shadow removal, and intrinsic image decomposition clearly demonstrate the effectiveness of this pre-training method. By working within the CroCo latent space to disentangle and swap the lighting latent vectors, there are many downstream photometric tasks that can be carried out purely through manipulation of the lighting latent. Our method provides a general-purpose lighting model that can easily be expanded with extra data or additional sub-modules.

**Reproducibility Statement**    For reproducibility, we describe the architecture of the model in detail throughout Section 3, along with task-specific model components for shadow removal and albedo estimation in Section 5.2. The datasets included in training data along with augmentations are discussed in Section 4 and the hyper-parameters are listed in detail in Appendix A. The code is publicly available at `https://github.com/alistairfoggin/CroCoDiLight`.

**Acknowledgments**    Model training was carried out on the Viking cluster, which is a high performance compute facility provided by the University of York. We are grateful for computational support from the University of York, IT Services, and the Research IT team.

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

## APPENDICES

## A HYPER-PARAMETERS

To train the single-view decoder, we trained for 120k iterations on ImageNet with a batch size of $48$ and a learning rate of $1 \times 10^{-4}$. The components $\mathcal{I}$ and $\mathcal{R}$ were jointly trained for 30 epochs (approx. 143k iterations) with a batch size of 12 and the same learning rate of $1 \times 10^{-4}$. The shadow removal and albedo estimation models $\mathcal{S}$ and $\mathcal{A}$ were both trained with a batch size of 32 and a learning rate of $5 \times 10^{-5}$. Due to different task-specific dataset sizes, they were trained with a different number of epochs. $\mathcal{S}$ was trained for 120 epochs (approx. 18.8k iterations), and $\mathcal{A}$ was trained for 40 epochs (approx. 27.5k iterations).

## B CROCO DECODING RESULTS

To demonstrate the necessity of training our own single-view decoder instead of using the CroCo v2 binocular decoder, we visualise the results of using each to encode and decode a single image. To use the binocular encoder as a single-view encoder, we feed the source image in twice. The CroCo

patches are passed in as both the primary and secondary view. Figure 7 shows the lack of fidelity in the CroCo decoder when compared to our single-view decoder.

| Input | CroCo | Ours | Input | CroCo | Ours |

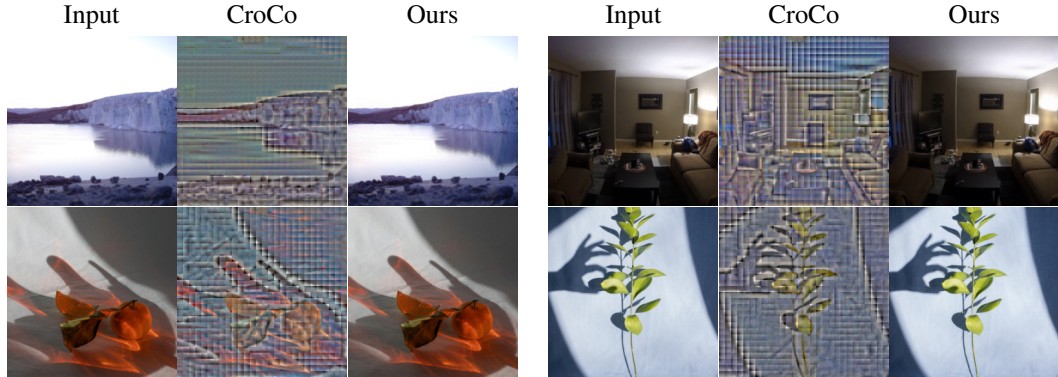

Figure 7: Comparison of the CroCo v2 binocular decoder compared to our single-view decoder.

## C  LIGHTING LATENT COMPRESSION

Figure 8 demonstrates how much lighting information is compressed and encoded into the lighting latent. To demonstrate this, the intrinsic latent patches are extracted from a blank black image, and then relit with the lighting latent from a reference image, before being decoded. This demonstrates the components of the reference image that are embedded into latent space. We also demonstrate the extent of information in the intrinsic patch embeddings by taking them from the reference image and relight it with a lighting latent consisting of only zeros ensuring the only information shown is from the intrinsic patches.

| Blank | Reference | Lighting | Reference | Intrinsics |

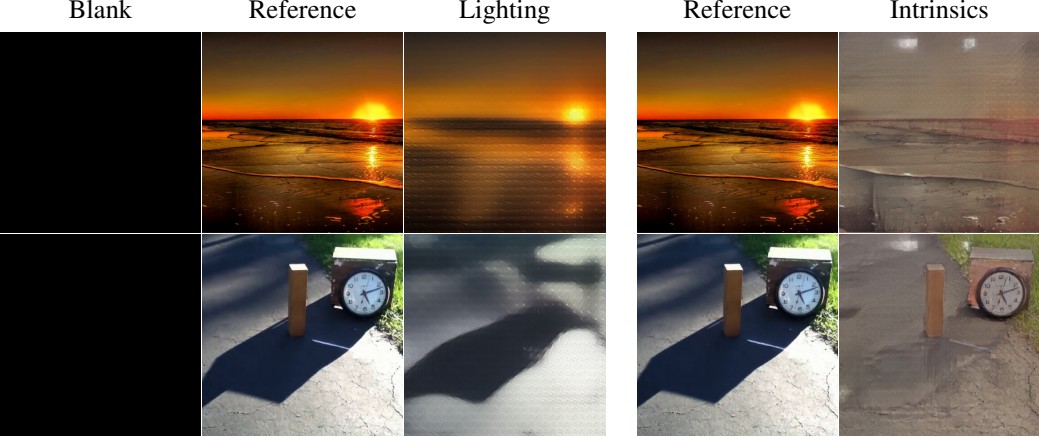

Figure 8: Visualisation of information encoded into the lighting latent and intrinsic patches. Intrinsic patches are disentangled from the blank image, and relit using the lighting latent disentangled from the reference image, and decoded to produce the lighting image. The intrinsics of the reference image are then entangled with a lighting latent vector that is all zeros before also being decoded to produce the intrinsics image.

## D  RELIGHTING EXAMPLES

Here we provide further examples of manipulating the lighting latent. In Figure 9, we take the lighting from one frame of a timelapse, and relight the rest of the frames to match it. We also demonstrate the opposite effect in Figure 10 where we freeze the intrinsics of a single frame and relight it to match every other frame. Finally we include a couple more examples of linearly interpolating the lighting latent in Figure 11. We extract the lighting latents per-tile from two frames, interpolate between

them, and relight the intrinsics of the first with the interpolated latents. Table 1 shows quantitative results of upsampling by interpolating halfway between every 7th frame in timelapses. A video demonstration of various lighting latent manipulations is included in the supplementary material.

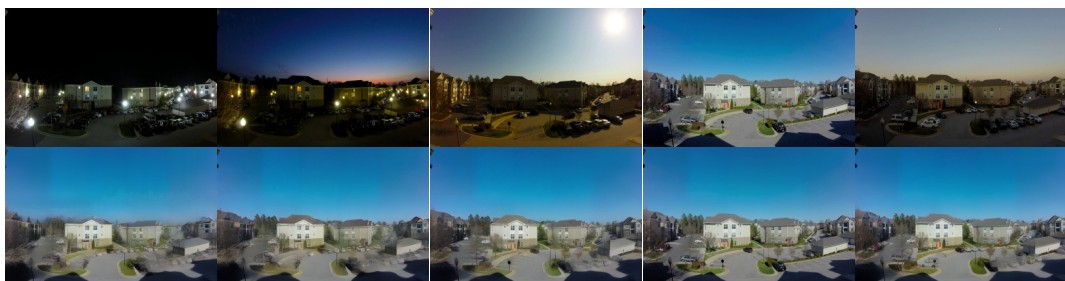

Figure 9: An example of extracting the lighting from the fourth column and relighting the intrinsic patches of the other frames to match it. The first row is of the ground truth input images, and the second row is of them relit accordingly.

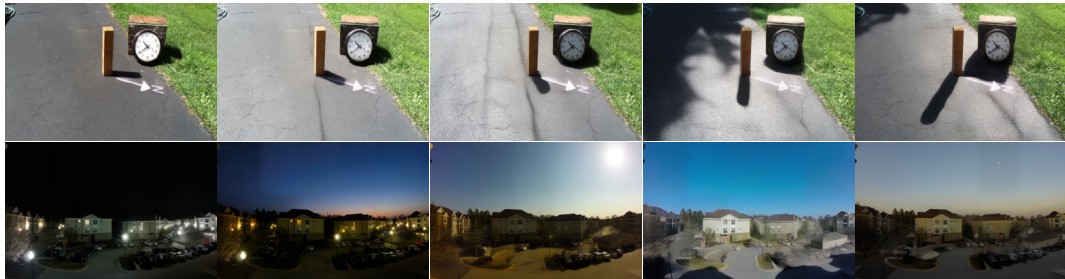

Figure 10: Examples of taking the intrinsic patches from one of the frames and relighting it with the lighting from other frames. These examples demonstrate preventing the clock from turning (which it does in the original frames), and keeping the same parked cars despite them changing throughout the day. The first row uses the intrinsics from the first column, but the second row uses the intrinsics from the second column due to a lack of detail in the first column.

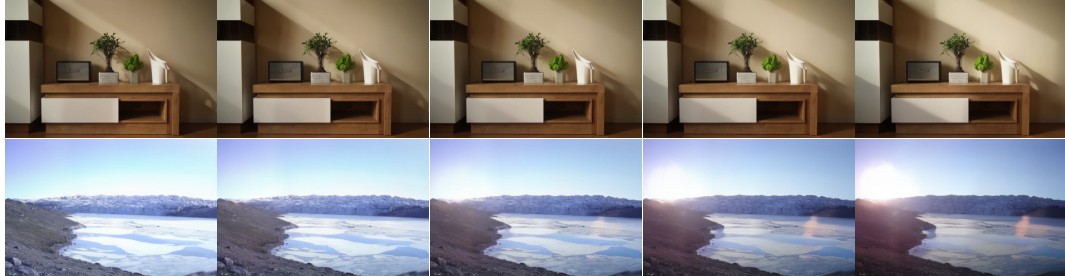

Figure 11: Further examples of the capabilities of linearly interpolating between lighting conditions in the lighting latent space.

## E    ALBEDO ESTIMATION RESULTS

To more clearly demonstrate the effectiveness of the albedo estimation module, we include several examples of albedo estimation from the IIW dataset in Figure 12. We extract $\hat{s}$ from the image, feed it through $\mathcal{A}$ to produce the albedo latent $s'_0$ which is used to relight the intrinsic patches.

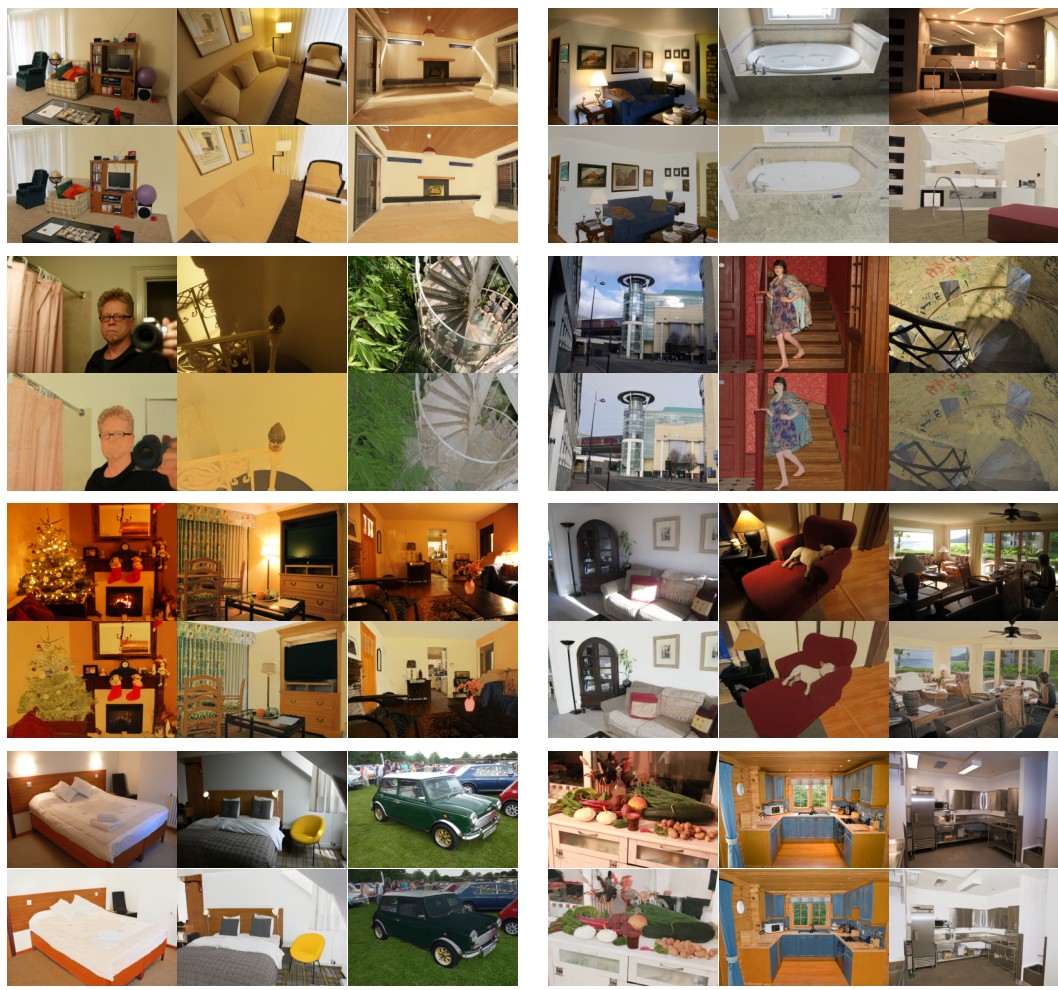

Figure 12: Additional examples of albedo estimation on IIW.

## F    SHADOW REMOVAL RESULTS

We also provide additional shadow-removal results on the SRD, ISTD+, and WSRD+ datasets in Figures 13, 14, and 15. This demonstrates the effectiveness of our model at shadow removal, seemingly in contrast to the metric results in Table 2. To explore the reason for this, we compare specific examples of shadow removal against the results of previous methods. Figure 16 shows key examples where our overall image consistency of shadow removal is better than other methods. The difference maps highlight that the parts of the image outside of the shadow have an overall colour shift with a greater difference than the other methods, but the difference in the shadowed part of the image is much less significant. This highlights that with improved colour mapping, our method could get a boost in the metrics causing them to more clearly reflect the shadow removal effectiveness.

### F.1    FAILURE CASES

While our method is generally effective at shadow removal as seen in the qualitative results, our model still has some failure cases as shown in Figure 17. Some challenging scenarios cause it to not remove the shadow. There are also failures caused by the use of the the sliding window. If a tile is completely covered in shadow, it does not have a big enough perceptual field to determine the necessary brightness and colour of the shadow-free regions. There are several potential ways of fixing this issue. Poisson image editing (Pérez et al., 2003) can be used to combine the tiles using their gradients to match a reference tile. The entire image can be resized to a single tile

Input  GT  Ours    Input  GT  Ours

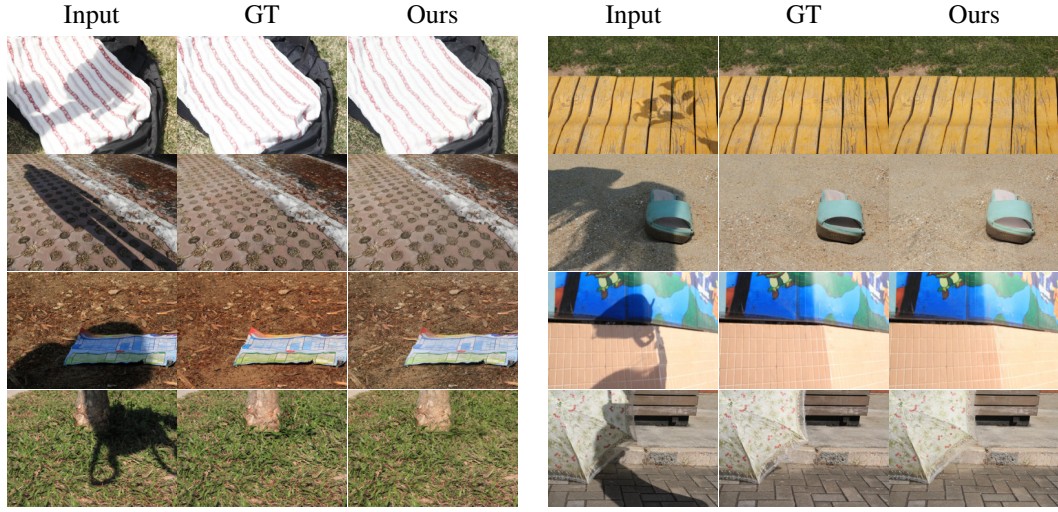

Figure 13: Additional examples of our shadow removal on SRD.

Input  GT  Ours    Input  GT  Ours

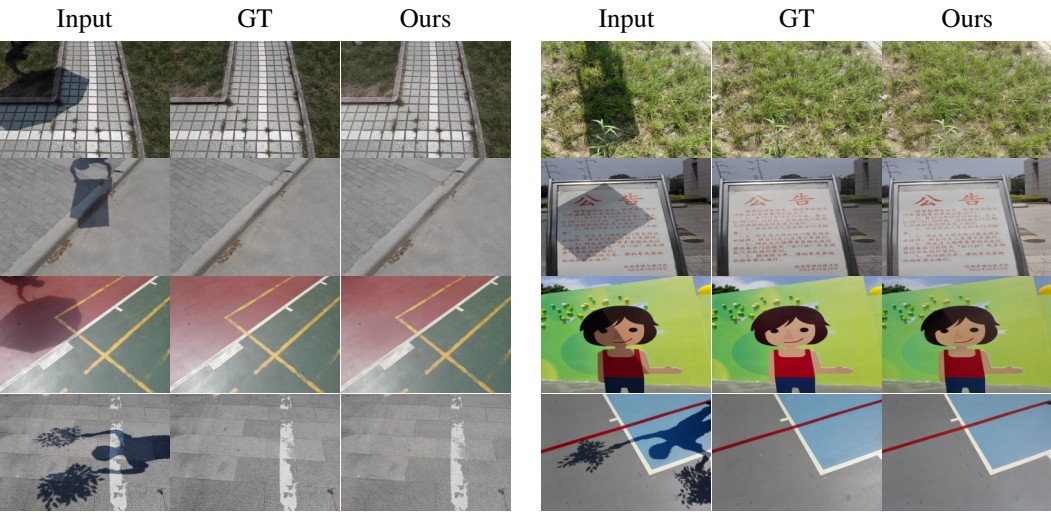

Figure 14: Additional examples of our shadow removal on ISTD+.

Input  GT  Ours    Input  GT  Ours

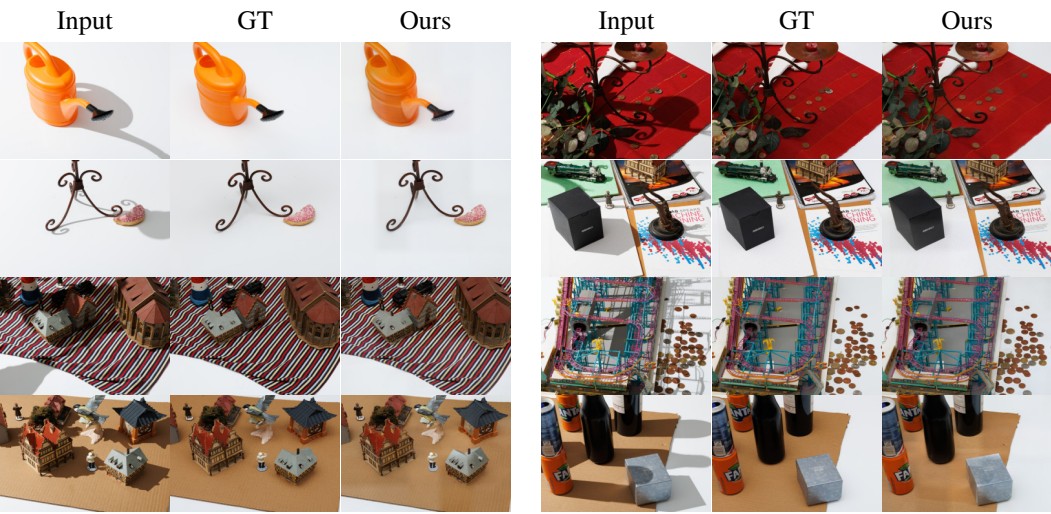

Figure 15: Additional examples of our shadow removal on WSRD+.

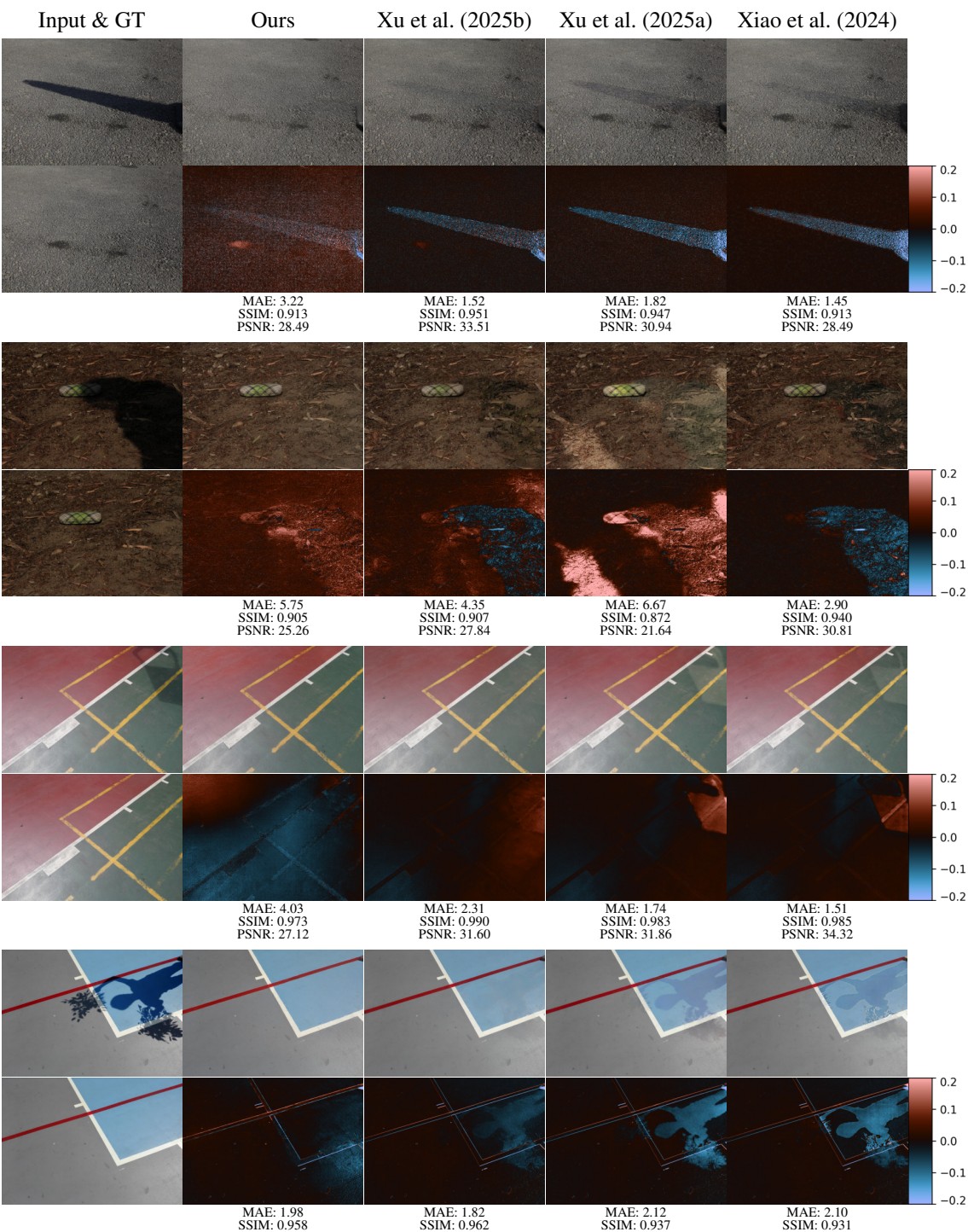

Figure 16: Some examples of shadow removal when compared to other methods. We show that our method produces effective results, while demonstrating a subtle overall colour shift that is detrimental to the metrics. Each pair of rows show the outputs along with a signed heatmap of the difference between the outputs and the ground truth averaged across channels and scaled up to be more visible. We also specify the metrics for each image. From left to right, the other methods we compare against are StableShadowRemoval, OmniSR, and HomoFormer.

which can be processed and upscaled to be a reference for adjusting the colour and brightness of the high-resolution tiles. Other methods may also be possible to allow self-attention between multiple lighting latents per-tile, or to allow a model to adjust the lighting latents based on the entire image.

| Input | GT | Ours | Input | GT | Ours |
|---|---|---|---|---|---|

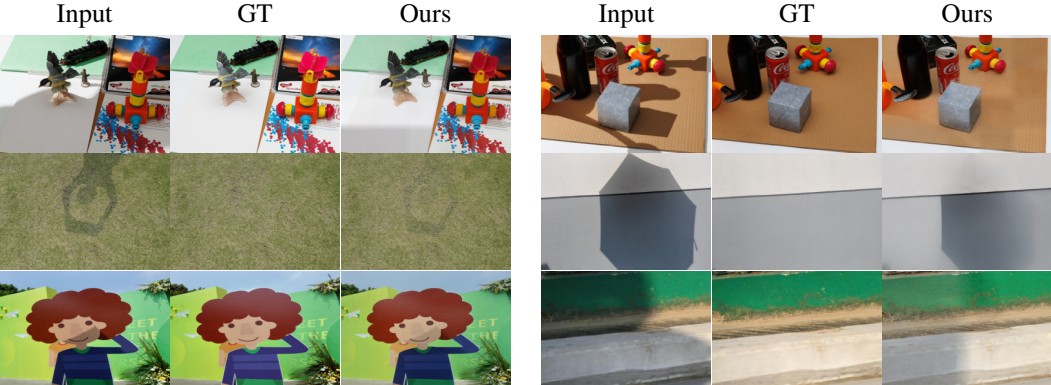

Figure 17: Shadow removal failure cases. Such failures include varying colours in tiles due to the sliding window having a small perceptual field, whereas others just fail to remove the shadows.

### F.2 Non-Failure Cases

During evaluation, we noticed that certain shadow-removal results seemed to be effective in ways that could not be accounted for by the metrics. Figure 18 demonstrates a couple of these examples. The method is able to distinguish between reflections and shadows in such a way that the shadow can be removed while retaining the reflection. This however is detrimental to the metric as the ground truth shadow-free images do not have some of the reflections. Another example of it doing better than the ground-truth is where it removes additional shadows that were not accounted for. Finally, we also noticed that when removing direct shadows in the WSRD+ dataset, it replaces them with ambient occlusion rather than removing it altogether. This fits with incorporating WSRD+ into training due to its design of turning an external light on and off, while the other shadow-removal datasets would not have this bias as they cast external shadows.

| Input | GT | Ours | Input | GT | Ours |
|---|---|---|---|---|---|

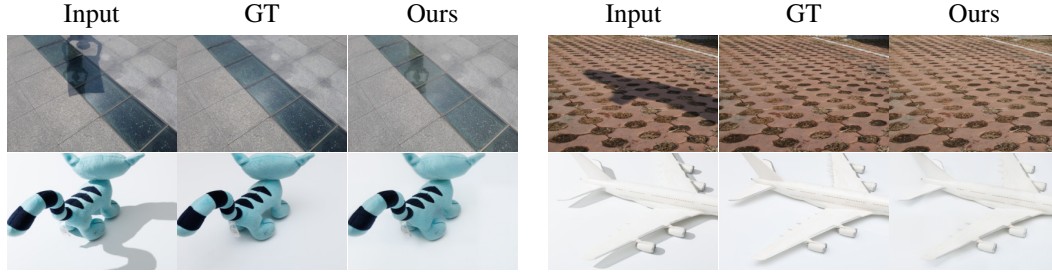

Figure 18: Shadow removal cases that may not match the ground truth shadow-free image, but are working better (i.e. removing additional shadows or not removing reflections). We also include cases where it replaces direct shadows with ambient occlusion.

### F.3 INS Dataset

StableShadowRemoval and OmniSR make use of the synthetic INS dataset created by Xu et al. (2025a) in OmniSR to improve results. Although our original model did not include it in training, we trained our models again with it (while keeping all previously used datasets) to see if there were any improvements. We adjusted the epochs to account for the change in dataset size to ensure comparison models had been trained for approximately the same number of iterations. Table 5 demonstrates a slight improvement on most benchmarks when it is included in the training process for $\mathcal{I}$ and $\mathcal{R}$, but including it in training of $\mathcal{S}$ was detrimental. It is worth noting that some cases it allowed for removal of shadows not accounted for in the benchmarks which is part of the cause of worse results (see Figure 19). Table 6 demonstrates that including INS in the training of $\mathcal{I}$ and

$\mathcal{R}$ was detrimental to albedo estimation. It was due to this and minimal improvements in shadow removal that we did not include it in training our final model.

| Ablations | | ISTD+ | | SRD | | WSRD+ | | INS | |
|---|---|---|---|---|---|---|---|---|---|
| INS in $\mathcal{I}$ and $\mathcal{R}$ | INS in $\mathcal{S}$ | SSIM ↑ | PSNR ↑ | SSIM ↑ | PSNR ↑ | SSIM ↑ | PSNR ↑ | SSIM ↑ | PSNR ↑ |
| | | 0.929 | 30.17 | 0.931 | 30.01 | 0.922 | 25.30 | 0.869 | 21.87 |
| ✓ | | **0.934** | **31.23** | **0.932** | **30.55** | **0.924** | **25.31** | 0.918 | 25.27 |
| ✓ | ✓ | 0.932 | 31.07 | 0.927 | 29.38 | 0.914 | 24.49 | **0.934** | **28.30** |

Table 5: Ablation study on the effect on shadow removal of including the synthetic INS shadow removal dataset in training the main model components $\mathcal{I}$ and $\mathcal{R}$, and in training the shadow removal module $\mathcal{S}$.

| | IIW | |
|---|---|---|
| INS in $\mathcal{I}$ and $\mathcal{R}$ | WHDR ↓ | WHDR (+0.5) ↓ |
| | **15.4**% | **14.3**% |
| ✓ | 17.6% | 14.9% |

Table 6: Ablation study on the effect on albedo estimation of including the synthetic INS shadow removal dataset in training the main model components $\mathcal{I}$ and $\mathcal{R}$.

| Input | GT | Ours - INS |
|---|---|---|

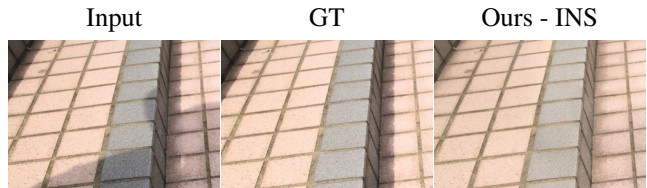

Figure 19: Example of the model where $\mathcal{I}$, $\mathcal{R}$, and $\mathcal{S}$ were all trained on INS correctly removing shadows that SRD did not account for in the shadow-free image.

