# OpenReview forum: "CroCoDiLight: Repurposing Cross-View Completion Encoders for Relighting"
_ICLR.cc/2026/Conference — ICLR 2026 Poster_

### Official Review · Reviewer_Zxsi · 2025-10-24

**Soundness:** 3
**Presentation:** 2
**Contribution:** 3
**Rating:** 4
**Confidence:** 3

**Summary:**

The paper proposes CroCoDiLight, which repurposes the pre-trained CroCo encoder for photometric tasks. The authors hypothesize that CroCo implicitly learns lighting information through cross-view completion training on image pairs with varying illumination. The method is demonstrated on tasks including lighting stabilization in timelapse, temporal upsampling, shadow removal, and intrinsic decomposition, trained on datasets two orders of magnitude smaller than CroCo's original training data.

**Strengths:**

- **S1.** Novel insight that cross-view completion models implicitly learn photometric understanding. The hypothesis that CroCo must estimate and manipulate lighting to complete masked patches across views with varying illumination is interesting and well-motivated.

- **S2.** Efficient learning paradigm requiring datasets two orders of magnitude smaller than original CroCo training. This demonstrates that photometric knowledge is already embedded in the pre-trained encoder and only requires extraction rather than learning from scratch.

- **S3.** Demonstrates feasibility of repurposing cross-view completion foundation models for photometric tasks, opening a new direction for leveraging geometric pre-training for appearance-related downstream applications.

**Weaknesses:**

- **W1.** My main concern is the paper's positioning and the scope of investigation. The paper is framed as an application showcase (e.g., lighting stabilization in timelapse, temporal upsampling, shadow removal, intrinsic decomposition, etc.), but its core contribution is the insight into repurposing foundation models. It would be much stronger if repositioned as a systematic investigation (similar to Probe3D) into which and how pre-trained vision foundation models capture photometric properties, and why. The current study is confined to CroCo, missing a crucial comparative analysis against other foundation models. An investigation should include:
    - Other two-view encoders (e.g., DUST3R, MAST3R) and matchers (e.g., RoMa, GIM).
    - Single-view models known for strong correspondence (e.g., DINOv2, DINOv3).
    - Multi-view models where two-view is a special case (e.g., VGGT, Pi-3).
  - Such a comparison would provide more generalizable insights into how different pre-training objectives (cross-view, contrastive, etc.) contribute to learning photometric understanding.

- **W2.** Mixed results on quantitative evaluations and missing some evaluations.
    - We only have quantitative results for shadow removal (Table 1) and intrinsic decomposition (Table 2), while the other applications (lighting stabilization in timelapse, temporal upsampling) lack any quantitative benchmarks.
    - The intrinsic decomposition results are state-of-the-art (Table 2), while shadow removal is not (Table 1). This is acceptable if the paper is repositioned as an investigation (per W1), where the goal is demonstrating feasibility rather than beating every SOTA. However, under the current paper's narrative, it is difficult to justify the advantage of CroCoDiLight over other specialized methods.
    - That is, if we shift the paper's focus from application showcase to systematic investigation, we don't need to provide quantitative results for all applications, and we don't necessarily need to beat every SOTA.

Overall, the paper's insight (cross-view completion models implicitly learn photometric understanding) is valuable, but the paper's positioning and scope are not strong enough to provide a comprehensive investigation of this insight. I welcome the authors' response to address these concerns.

**Questions:**

- **Q1.** (Related to W1, authors may answer together) Have you experimented with other two-view foundation models like DUST3R or MAST3R (which build on CroCo)? What about single-view and multi-view models?

- **Q2.** (Related to W2, authors may answer together) Could you provide quantitative metrics for lighting stabilization and temporal upsampling tasks? For example, comparing temporal coherence metrics or perceptual quality against baseline interpolation methods? Or is that something beyond the scope of this paper?

---

> ### Author Response · Authors · 2025-11-20
>
> Thank you very much for your valuable review and insightful suggestions!
>
> 1. **W1 and Q1 - Other encoders and paper scope**: We would first like to direct you to our general response regarding why we focus on CroCo, which we reiterate here. CroCo has a definitive reason as to why it should have learnt lighting information as it is supervised to fill in relit patches from a secondary view. We have not tested models such as DUST3R and MAST3R, because we reason that while they have been trained from CroCo, their fine-tuning on 3D may have reduced the relighting capabilities as they focus on being *invariant* to lighting changes in their 3D output.
>
>     With the proposal of carrying out a systematic analysis of foundation models, we recognise the value of such a study, but also argue that papers such as Probe3D do such a study built on the foundations of previous papers like “Repurposing Diffusion-Based Image Generators for Monocular Depth Estimation”. Such papers demonstrate the feasibility of extracting 3D from a single foundation model, which our paper does with extracting photometric information from CroCo.
>
>     We also argue that our method is not so much an application showcase, but rather making use of such applications to demonstrate the photometric information that has been extracted from CroCo latent space. With the evidence shown in our paper, future studies can be done to evaluate photometric information embedded in other foundation models.
>
> 2. **W2 and Q2 - Quantitative evaluations**: Lighting stabilisation and interpolation are primarily demonstrative of the extent of the lighting latent space extracted from CroCo latent space. Lighting stabilisation is a unique application that does not have any baseline with which one can evaluate it. For lighting interpolation, as our method does not do arbitrary video temporal upsampling but rather fixed-view upsampling with limited structural dynamics, standard evaluations such as Vimeo-90k cannot be run. It demonstrates the extent to which lighting has been extracted into latent space. The video in the supplementary material shows perceptually smoother upsampling results than cross-fading. We did run some qualitative evaluations of temporal upsampling which we now show in **Table 6**. As the focus is on evaluating perceptual performance of temporal upsampling, we evaluated a video-specific metric rather than per-frame metrics. For our interpolations we sampled every 7th frame of 3 separate timelapses, and interpolated halfway between each frame in both lighting latent space and in image space for comparison. We then ran FloLPIPS for both methods on image triplets of the reference frames on either side and the interpolated frame with the reference ground truth frame. Our lighting latent space interpolation shows improvement over image space interpolation on two of the timelapses and similar results on the day-night cycle. As our interpolation is in lighting latent space, we keep the intrinsics from the start of each triplet, meaning that our method does not handle structural dynamics. We reason that this is why the day-night cycle has slightly worse results than cross-fading which can somewhat blend structural dynamics.
>
>     When it comes to the shadow removal metrics and the disparity between the quantitative and qualitative results, we refer again to our general response. Other methods fine-tune their models specifically on each benchmark for the best results, but our method **trains on all the datasets to generalise and remove bias** inherent in a single dataset. In **Figure 15** we provide visual results and image-specific metrics comparing against other methods where you can clearly see that ours removes the shadows leaving a slight global colour-shift while other methods have distinct shadow artefacts.

---

### Official Review · Reviewer_3fbg · 2025-11-01

**Soundness:** 3
**Presentation:** 3
**Contribution:** 3
**Rating:** 8
**Confidence:** 3

**Summary:**

The authors introduce a method to disentangle CroCo latent representations into two components: a global latent vector capturing illumination, and patch-wise latent vectors representing the intrinsic properties of the scene. The model is trained in a self-supervised manner using pixel-wise aligned image pairs taken under different lighting conditions, guided by per patch cross-lighting and intrinsic consistency losses. They demonstrate that the disentangled latent space can be effectively leveraged for novel tasks such as interpolating between lighting conditions, shadow removal, and albedo estimation.

**Strengths:**

I found the approach of using an encoder-decoder architecture inspired by the Croco architecture to disentangle illumination from intrinsic scene representation both interesting and original. The design of the self-supervised training framework, particularly the proposed losses, appears well thought out and conceptually sound.

The paper demonstrates, through a range of tasks—including lighting interpolation, shadow removal, albedo estimation, and intrinsic image decomposition—that the proposed disentanglement approach is effective. The latent representations prove useful for handling these diverse downstream tasks. While the model does not outperform state-of-the-art methods specifically tailored for each task, the results are nonetheless promising, and the visual examples are rather convincing.

**Weaknesses:**

It is probable that leveraging the pretrained CrocoV2 model is beneficial, because the model was trained on a large set of image pairs captured under varying lighting conditions. Additionally, photometric augmentations applied during training likely enhanced the model to be robust to lighting changes. Still the experiments in the paper do not completely prove this.  How would the model work if instead of the Croco encoder the  MAE, DINOv2 or DINOv3 encoder is used and disentangled? Would the model perform less well on the downstream tasks?

Also, the ablation in Table 6 raises a concern: the architecture of the two models compared are no longer identical so while supposedly it is true, it is not directly shown that the gain comes from the pre-trained model and not from the architecture choice.  It would have been insightful to evaluate also a model that retains the CroCo encoder architecture but is initialized from scratch. While indeed the training dataset is smaller, the learning task seems simpler than the hidden content reconstruction, making such an experiment worthwhile. Note also that the performance gain with the CroCo pretraining is significant only for the Intrinsic Image Decomposition task, much less for the Shadow Removal.

**Questions:**

I like the illustration and the narrative in Figure 1 as it effectively conveys how the CroCo model implicitly learns to extract content information from the second image and appearance information from the first, guided by the training data. This raises an interesting question: could the two models be integrated to jointly learn both the reconstruction and the disentangled latent representations assuming relevant training set (e.g. image triplets)? Such a unified approach—where mask content reconstruction and disentangled latent representations are jointly learned—could potentially enhance consistency and improve performance not only on the downstream tasks explored in this paper (e.g., shadow removal, albedo estimation, lighting interpolation), but also on geometric tasks such as 3D reconstruction.

---

> ### Author Response · Authors · 2025-11-20
>
> Thank you very much for your review, and the thought and effort that you put in!
>
> 1. **Alternative Encoders**: In our general response we highlight why we focused on just CroCo and not alternative encoders. To reiterate the points we made there, CroCo has a much stronger reason as to why it will have learnt lighting and be capable of relighting. Other models are likely to have learnt *invariance* to lighting, but CroCo explicitly has to cross-project and relight patches from the second image onto the first masked image. While there is potential to use our architecture with other encoders, our main focus is demonstrating that CroCo has inherently learnt lighting information that can be extracted. It fits with other similar papers repurposing foundation models such as “Your Diffusion Model is Secretly as Zero-Shot Classifier” which demonstrate proxy information has been learnt by individual models and can be extracted.
>
> 2. **Ablation with Full CroCo Architecture**: Thank you for raising concern with regards to the ablation study. As mentioned in our general response, we are currently carrying out the ablation evaluation with an identical architecture in order to have a fairer evaluation. We will notify you when the training and evaluation is completed.
>
> 3. **Potential for unified training**: That is an interesting point you raise about unifying masked cross-view reconstruction and disentangling lighting latent vectors. With a specialised dataset such a method may be possible but the volume of data available for training would drastically decrease making it a much more challenging task. It would likely lead to a more complex training method and architecture to attempt it either with that kind of data with limited volume, or to make use of existing image pair datasets. However there is potential to explore such a line of thinking.

---

> > ### Author Response · Authors · 2025-11-24
> >
> > **Update on Ablation Study**: We have completed our ablation evaluation using an identical architecture with the full CroCo encoder and our decoder. As shown in **Table 3**, it demonstrates that when using an identical architecture, the pre-training provides a significant improvement proving our hypothesis regarding the photometric information embedded in CroCo. The simplified model does help when training from scratch due to the reduced number of parameters, but our full method still outperforms both ablations.

---

### Official Review · Reviewer_FRcB · 2025-11-01

**Soundness:** 3
**Presentation:** 3
**Contribution:** 3
**Rating:** 4
**Confidence:** 4

**Summary:**

This work presents CroCoDiLight, which leverages supposedly inherent lighting disentanglement capability within CroCo to modify & repurpose CroCo for de-lighting / relighting tasks. To achieve this, the authors introduce two networks that explicitly separate CroCo's patch embeddings into a lighting vector and lighting-invariant latents, then recombine them, demonstrating that photometric understanding is already embedded in CroCo's representations and can be efficiently extracted for explicit control and various relighting tasks such as interpolation between lighting conditions, shadow removal, and albedo estimation.

**Strengths:**

- The paper is well-written and easy to follow. Good writing.
- The paper starts with a strong observation & hypothesis, recognizing the (possible) inherent capability within the original CroCo paper that its encoder implicitly encodes illumination information, which enables delighting & relighting during its novel view reconstruction task, and extending it to the hypothesis that this capability can be explicitly harnessed to achieve photometric tasks such as relighting/shadow removal/delighting. I believe the work is well-motivated and tackles an interesting question about the nature of CroCo and its representation.
- The authors offer a simple and intuitive solution to the problem (though this might point to the lack of novelty, as I would mention in the weakness section) by including a latent vector that disentangles lighting from geometry during the training phase. The method is simple and straightforward, effectively achieving its goal of lighting disentanglement as shown in the results.

**Weaknesses:**

- The original CroCo paper was a representation learning paper, focused on pre-training the model to be generally more suitable for various 3D / NVS downstream tasks from a simple two-view reconstruction loss. However, it seems that this work is more focused on training a model towards each specific downstream task (relighting / shadow removal / intrinsic image decomposition), which makes this work more closely aligned with existing relighting methods, of which there are already many. However, in this view, the core method of this paper (adding a separate latent vector for style and teaching model to change 'style (lighting)' of the image) very closely resembles previous GAN methods that achieve similar goals in generative scene and seems to lack novelty. Is there a more general implication for this method that may be relevant to representation learning / other downstream tasks, as was the original CroCo?
- The method requires datasets with identical geometry under different lighting, significantly limiting available training data. While synthetic datasets like HyperSim could be used, they introduce domain gap issues. How the authors address this fundamental limitation remains unclear.
- Lighting manipulation requires "walking" through latent space, making it difficult to achieve specific desired lighting conditions. The paper lacks a demonstration of how users can specify target lighting or achieve reproducible, controllable results without another scene that has desired lighting - can this point be further elaborated?
- What does this method have in advantage in comparison to Diffusion-based relighting methods, especially IC-Light (ICLR 2025), whose lighting can be controlled with text prompt as well as can be applied to various domains beyond scene imagery (i.e. including human faces, etc.)? Please elaborate.

**Questions:**

Please see Weaknesses section.

---

> ### Author Response · Authors · 2025-11-20
>
> Thank you very much for your thorough and valuable comments in your review!
>
> 1. **Representation learning versus downstream task engineering**: We see our paper as more in the direction of representation learning and would like to clarify a possible misunderstanding. The delighting transformer (I), relighting transformer (R) and decoder (D) are all trained generically in order to disentangle in CroCo latent space into a global lighting latent and per-patch latents representing intrinsic scene properties. None of these models are further trained for downstream tasks. The downstream tasks either use the disentangled representation directly (interpolation, lighting stabilisation and temporal upsampling) or require only a model to translate the global lighting latent (for albedo estimation and shadow removal). These translation models, S and A for shadow removal and albedo estimation respectively, are trained just to manipulate the lighting latent space to achieve those tasks using the frozen components I, R, and D. We realise the wording of how S and A are initialised may have given the impression that we fine-tune R which we do not do. They are created by making a copy of R, and trained to learn to transform the lighting latent s0 into the required latent for the task, but R itself is left unmodified in the task-specific stages. We have now clarified the wording in the paper and added **Figure 19** to show the task-specific training process.
>
> 2. **Relationship to GAN-based methods**: We agree that much prior work, including GAN-based methods, also learns a latent space for lighting. However, the focus of our method is not to generate new arbitrary lighting conditions either randomly or with conditioning, but rather to take real images and to extract the lighting that exists into the latent space. This lighting latent vector can then be manipulated for specific tasks, transferred to another image from the same viewpoint which has a different lighting condition, or interpolated to produce intermediate lighting conditions between two images. The representation learning is for understanding the existing lighting rather than generating arbitrary lighting.
>
> 3. **Requirement for fixed view/different lighting training data**: The main argument of our paper is that the CroCo pretraining on different view/different lighting data means that we need very little fixed view/different lighting data for our training. As you say, synthetic data introduces a domain gap issue so we focus on real datasets such as timelapse or artificially changed lighting pairs for training. We do use some synthetic training data but this is primarily to ensure that our model learns a lighting latent space that includes a subspace that corresponds to entirely delit (i.e. albedo only) images. There is certainly scope to increase the available real data. We made use of BigTime, which is timelapse data, but is not a large dataset by today’s standards. There is plenty of room for it to be expanded to incorporate much more data. The main constraint on timelapses is to have limited structural dynamics visible, but this could be mitigated by incorporating masks of dynamic objects. We can also expand the domain of our model to better handle human faces by including face lighting datasets such as those collected on a light stage. But even without expanding, we have already shown that existing data (that is two orders of magnitude less than that needed to train CroCo) is sufficient for the results we obtain.
>
> 4. **Pros/cons versus diffusion relighting**: Approaches such as IC-light are solving a different problem. Text-based lighting conditioning provides a natural, but imprecise, means to edit foreground lighting and hallucinate background content. Background image-based relighting changes background scene structure. On the other hand, our approach provides an explicit disentangling into intrinsic scene structure and a navigable lighting latent space. Lighting latents are exactly defined by an image meaning our representation is ideally suited to interpolation or timelapse applications. Our relighting results exactly preserve scene structure since intrinsic patches are kept constant and only the global lighting latent is changed. So our goal is to fix the scene and change the lighting present in the scene whereas IC-light is creating an image with a new background in which foreground content is harmonised.
>
> 5. **Walking through latent space**: The previous answer hopefully partially answers this. In our representation, lighting is specified by images - e.g. to provide the start and end of an interpolation. We do not tackle applications that allow a user to provide, for example, text conditioning. The two exceptions are shadow removal and albedo estimation where we essentially learn to convert a given lighting condition into another one (that either removes shadows or removes all lighting to provide an albedo-only image).

---

### Official Review · Reviewer_qx9H · 2025-11-03

**Soundness:** 2
**Presentation:** 3
**Contribution:** 2
**Rating:** 4
**Confidence:** 4

**Summary:**

This paper explores whether CroCo encoders, originally trained for cross-view completion with geometric objectives, also implicitly learn photometric representations due to illumination variations in training pairs. The authors propose CroCoDiLight to make this knowledge explicit through a delighting transformer that disentangles CroCo patch embeddings into a single lighting latent and intrinsic patch latents. Also a relighting transformer R that recombines them, and a single-view decoder D for high quality RGB reconstruction. Training uses only 57k pixel-aligned image pairs with different illumination, two orders of magnitude less than CroCo's training data as claimed in the paper. The method demonstrates applications in lighting interpolation, timelapse stabilization, shadow removal, and albedo estimation. Results show state-of-the-art on IIW among methods not trained on IIW, though shadow removal and construction metrics are good but not convincing.

**Strengths:**

The hypothesis that CroCo learns implicit photometric understanding is intuitive and the paper validates it convincingly. The delight-relight framing is elegant.
Needing only 57k pairs versus CroCo's 5.3 million is a strong practical advantage, especially given that aligned multi-illumination data is scarce. The paper shows strong albedo results- achieving 14.3% WHDR on IIW without training on IIW is impressive and suggests the intrinsic latents capture meaningful scene properties.
The paper covers multiple downstream tasks and provides extensive qualitative results. The failure case analysis in Appendix F is honest and valuable. The timelapse stabilization and lighting interpolation demos are compelling and showcase practical utility though there are some limitations such as shadow motion not being entirely smooth and the method struggling with sharp shadow boundaries that move rapidly across scenes.

**Weaknesses:**

Sharp shadow handling (shading effects, cast shadows, etc) is inadequately addressed: This is my biggest concern. The method uses a single global lighting latent, which fundamentally cannot capture the geometric information needed for sharp shadow boundaries. Sharp shadows arise from point lights and hard occluders, they encode precise light direction, occluder position, and surface geometry. A single image-space vector cannot represent this information, especially when shadow boundaries need to move correctly across multiple frames. The evidence is scattered throughout:

Fig. 17 shows direct shadows being replaced with ambient occlusion
Section 5.1 notes shadow motion during interpolation is "not entirely smooth" and Section F.1 admits tiles fully in shadow fail.
Most successful examples (Figs. 3-4, 8-10) show soft shadows, diffuse lighting, or outdoor scenes with gradual illumination changes
The timelapse examples work well for slow sun movement creating soft shadow transitions, but would likely fail for a person walking past a lamp creating sharp moving shadows

Tiling artifacts is unresolved: Section 3.5 and Fig. 16 show color inconsistencies from the sliding window approach. Paper mention potential fixes (Poisson blending, global reference tile) but don't implement them. Why present solutions but not evaluate them? The lighting latent being "optimally used" at 448×448 is a fundamental design limitation. This significantly undermines the high-resolution claims. Shadow removal metrics don't really match qualitative results.

Fig. 17 shows cases where your method is "working better" by removing additional shadows, but this also suggests the model isn't learning what the benchmark defines as shadow removal in my opinion.

Limited architectural justification- Why a single lighting latent and why D=1024?

Paper didn't provide ablations on:

Multiple lighting latents per image/tile (which would help with local lighting)
Lighting latent dimensionality (is 1024 dimensions necessary? wasteful?)
Spatial lighting maps vs. single vector

The Table 3 ablation uses a much simpler baseline (just linear embedding + DPT head), making it unclear whether gains come from CroCo features or better architecture. A fairer comparison would use the same I/R architecture without CroCo pretraining.
Image-space lighting is a fundamental limitation: Section 5 and Appendix C mention the lighting latent works in "image space" not "world space" but don't explore the implications. This means:

The method can't handle viewpoint changes
It can't reason about 3D light positions or directions
It's brittle to even small camera motion
Shadows will appear in wrong positions if the camera moves slightly

**Questions:**

Can you quantify performance degradation as shadow sharpness increases? Even a simple analysis binning test images by edge gradient magnitude or manually annotating hard vs. soft shadows would help establish the method's scope.
Why not implement the color correction solutions you mention (Poisson blending, global reference) and show results? This seems critical for addressing both the metrics gap and the tiling artifacts.
A dimension ablation (dimensionality of S0) would help understand what information is being compressed.
For the world-space vs. image-space issue- did you try encoding light direction or position explicitly? Even rough geometric cues might help.
How does the method handle colored lighting vs colored surfaces? The disentanglement seems like it would be ambiguous
Lack of extensive ablation studies-
Multiple lighting latents per image/tile (which would help with local lighting)
Lighting latent dimensionality (is 1024 dimensions necessary? wasteful?)
Spatial lighting maps vs. single vector

---

> ### Author Response · Authors · 2025-11-20
>
> Thank you for your valuable and detailed review, and for the time that you put into it.
>
> 1. **Sharp shadow handling**: First, we would like to distinguish between *removal* of sharp shadows versus *synthesis* or reconstruction of sharp shadows. Our model is extremely capable at removal of shadows with sharp edges. For example, see Figures 5, 12, 13, 14 and 15. We see no difference in shadow removal performance related to the sharpness of the shadows and so don’t think breaking down results by this measure would be informative.
>
>     On the other hand, we agree that *synthesis* of shadows, as captured by our global lighting latent vector, does sometimes struggle to represent shadow boundaries precisely. For example, in Figure 9 top row, the shadows are all represented by the global lighting vector for each frame. Here, the shadow edges are slightly blurred and the shape of the overall shadow is slightly rounded. However, if you watch the video of this sequence supplied in supplementary material, what is clear is that the shadow edges move rather than cross-fade as in the linear interpolation approach. We would also contest the claim that this video shows “slow sun movement creating soft shadow transitions”. The clock casts sharp, well-defined shadows that transition in an almost binary manner between frames. When we downsample the frame rate for our temporal upsampling application, it exactly simulates dealing with input that has sharp, fast moving shadows.
>
>     As the reviewer correctly points out, the inability to synthesise sharp shadow edges is a consequence of using a single global latent vector to describe image space lighting. A requirement of our formulation is that the lighting representation must act as a bottleneck. If it is too powerful, the disentangling network can learn the trivial solution of representing the whole image in the lighting representation and setting the local intrinsic patches to zero. Our choice of 1024D is simply because this is the latent dimension of CroCo and we must match this in order for attention between lighting and CroCo patches to work. For this reason, we cannot run an ablation on the dimensionality of the lighting latent. We note that when our method is applied to larger images by tiling, some lighting locality is introduced since there is a 1024D vector per tile position.
>
>     To correct one misunderstanding, **Figure 17 is illustrating a quirk of a dataset, not a failure case of our shadow removal**. This dataset systematically swaps a point light source for ambient light. Hence, in the ground truth shadow-free images, cast shadows really are replaced by ambient occlusion from the diffuse lighting remaining (compare input and ground truth shadow-free images). Our latent translation network has learnt exactly this behaviour for these images: *remove* hard cast shadows and *synthesise* ambient occlusion. If this behaviour is undesirable, such training pairs only need excluding from the training set of the latent translation network.
>
> 2. **Quantitative shadow removal results**: We refer to the general answer given above but restate that the vast majority of shadow-removal methods fine-tune their model on the training split of each specific benchmark. Our method in contrast trained a general-purpose network that works purely in the lighting latent-space to remove shadows and it works across benchmarks. This is very similar to how we did our albedo estimation which you noted as a strength. We focus on generalisability across data rather than just hitting the best number on specific benchmarks. The result of training it on all the shadow-removal datasets as we did is that our model can remove shadows across benchmarks and it removes some of the bias of each dataset, which is a blessing and a curse. Our model is generalisable, but may not perform optimally for each specific benchmark by learning its bias. This is a limit of shadow removal benchmarks themselves as they each have different definitions and biases in how shadow removal should work, partially due to the challenge of getting ground-truth shadow-free images. WSRD+ shines a direct light to cast shadows from objects within view, and the shadow-free image is made with a diffuse overhead light which encourages the learning of ambient occlusion which our model also learnt. ISTD+ and SRD have externally cast shadows in sunlight, and pairs of images have slight misalignment in colour and pixel position. However, they sometimes have extra shadows or objects not caused by the dataset creators. If our model took shadow-masks into account it could have the potential to improve on those specific benchmarks, but that would also limit the applicability of our model in general.
>
> (continued in response below)

---

> > ### Author Response · Authors · 2025-11-20
> >
> > (continued from previous response)
> >
> > 3. **Tiling artefacts**: We acknowledge this as a weakness. Extending transformer-based models to arbitrary input size is a common challenge and other CroCo-derived architectures such as DUST3R also struggle with input outside their natural resolution. While more sophisticated blending strategies might better hide the tiling artefacts, the focus of the paper is on the core adaptation to CroCo while blending strategies is a more application-specific engineering step.
> >
> > 4. **Limitations of image-space lighting representation**: We acknowledge that this is a limitation and all of your observations about the implications of this are correct. But the simplicity of the representation is what makes it feasible to estimate given only the weak supervision we have. While changes in camera position could introduce shadow misplacement artefacts, we didn’t notice this as a significant problem. As an example, the glacier timelapse does include quite significant camera motion yet the lighting stabilisation results are still promising. We believe that a promising direction is to use the cross-view cross-attention in the CroCo decoder as a means to reason about view independent lighting but leave this to future work.
> >
> > 5. **Did you try encoding light direction or position explicitly?** We have not tried this. The necessary training data is even rarer than aligned pairs of images, and it is very task-specific. In our view, this would be a natural further decomposition of the global lighting latent but our current representation requires minimal supervision and avoids the fragility of incorporating any explicit 3D inductive biases into the representation.
> >
> > 6. **Coloured lighting vs coloured surfaces**: In general this is a fundamental ambiguity in intrinsic image decomposition or inverse rendering and other methods would be similarly uncertain, To our knowledge, the benchmark datasets do not include examples with coloured lighting. However, our global lighting representation may help here since regions of the image that disambiguate the lighting colour could help resolve the ambiguity elsewhere within the image.

---

> ### Author Response · Authors · 2025-11-24
>
> **CroCo Encoder Ablation**: We have now carried out a further ablation regarding the effectiveness of the CroCo encoder as we have now added to our general response. As shown in **Table 3**, when training the model with an identical architecture, it does significantly better with the pre-training than when trained from scratch. The simplified architecture still does well, but not quite as good as using the pre-trained weights of the full model. All of this indicates the substantial information embedded in CroCo latent space.

---

### Author Response · Authors · 2025-11-20

A number of comments were shared by multiple reviewers and so we respond to these collectively first.

1. **Shadow removal evaluation**: We do not believe that the shadow removal benchmark metrics properly reflect performance on the task for the following reasons: 1. Often the overall exposure or image brightness is different between input and ground truth output and this is systematic for each of the different benchmarks, 2. The task is ill-defined: sometimes there are shadows in the input image that are still present in the ground truth shadow-free image (or other phenomena like reflections are removed besides the shadows) and there is no way for a model to know which shadows are to be removed, 3. Often the ground truth image without shadow is not perfectly aligned with the input image so a lot of the error simply reflects geometric misalignment (this rewards methods that produce blurry output and does not reward complete removal of strong shadow boundaries). **We strongly encourage the reviewers to look at Figure 15** in the Appendix. The three very recent state-of-the-art methods against which we compare **leave clearly-visible shadow artefacts in their removal result** while our result leaves no discernible shadow. We have updated the visualisation of the signed-error-to-ground-truth maps to show them as heat maps with a colour bar. This illustrates that the shadow region is strongly visible in the error images for the comparison methods (indicating residual shadows have been left) while our errors are more uniformly distributed across the image. We have also added quantitative errors for each result. Even though our method does the best job perceptually of removing the shadow, it has the worst performance on almost every metric and image. We argue that this illustrates the weakness of the benchmark.

    We also emphasise that our approach is much more general than the specialised shadow removal methods we are comparing against here. Specialised methods train multiple shadow removal models fine-tuned on each individual benchmark. We learn a general disentangled latent space that can be used for a variety of downstream tasks (relighting, lighting interpolation, shadow removal, albedo estimation etc). The lighting latent translation network that we train for the task of shadow removal is general and not trained specifically for one dataset.

2. **CroCo encoder ablation**: Two reviewers suggested an alternative way in which we could have done the ablation that was intended to show the benefit of CroCo pretraining for relighting. With hindsight, we agree that their suggestion is a fairer way to do the comparison. We are re-running the ablation study where we use exactly the same architecture as in our full CroCoDiLight model but where the CroCo encoder is no longer pretrained. Once the evaluation is complete, we will update the paper with the results and post a comment to notify you.

3. **Why CroCo? Why not test many foundation model learnt representations like in Probe3D?** We answer this in two parts.

    First, CroCo has a particular property that motivates the whole paper. CroCo’s training objective requires it to explicitly generate relit patches. In other words, we already know that CroCo’s latent space can be decoded to RGB pixel space in a way that modifies lighting. While generic feature extractors like DINO, matchers like RoMa or 3D reconstruction models like VGGT will certainly learn *invariance* to lighting, there is no obvious reason that their latent space will learn to *modify* lighting nor that it is suitable for high quality decoding to RGB pixel space. DUST3R and MAST3R are themselves based on CroCo but with a training objective (3D reconstruction) that we expect would reduce rather than enhance relighting capability.

    Second, while Probe3D considers many different foundation models, it follows a long line of work showing that largescale models implicitly learn 3D understanding. We see our work more in line with precursors to Probe3D like “Repurposing Diffusion-Based Image Generators for Monocular Depth Estimation”. Here they show only that Stable Diffusion can be finetuned for monodepth estimation. While subsequent work like Probe3D tested whether this specific observation generalises to other backbones, this does not devalue the initial observation nor the usefulness of the model derived specifically from Stable Diffusion. We show the same here for CroCo while agreeing that testing the wider question on other backbones would be interesting future work.

---

> ### Author Response · Authors · 2025-11-20
>
> **Paper edits**: In light of your reviews and comments, we have made a couple edits within the paper to make some clarifications in our wording and figures:
>
> - Shadow removal - We highlighted how other methods fine-tune their models on specific benchmarks which achieves better quantitative results but reduces the generalizability. Our model on the other hand trains across datasets to enhance generalizability despite detrimental metrics. In many cases we have better perceptual outputs as shown in figure 15 which has also been updated to make the difference maps clearer as a signed heatmap, along with per-image metrics.
>
> - Task-specific training - We clarified our wording in how we train the components A and S components. We added figure 19 demonstrating the process.
>
> - Quantitative evaluation of temporal upsampling and interpolation - We have added table 6 with the quantitative results of temporal upsampling.

---

> > ### Author Response · Authors · 2025-11-24
> >
> > **Updated Ablation Study**: We have now completed the evaluation of training a model with an identical architecture to the CroCo encoder but from scratch. We made sure to use the same data, learning rate, and number of iterations. The results are in **Table 3** in the ablation study. Training an identical architecture from scratch significantly reduces performance. In other words, **the CroCo v2 pretraining of the encoder provides a significant boost to performance**. This strongly supports our argument that the CroCo v2 training objective and largescale different view/different lighting training data is key to enabling us to learn photometric tasks from a much smaller dataset of fixed view/different lighting pairs. The available fixed view/different lighting training data is too small to train such a large architecture from scratch.

---

> > > ### Author Response · Authors · 2025-12-02
> > >
> > > **Final Update on Paper Revisions**: We have made the final revisions to the paper. In particular, we have **reordered the figures** for clarity and logical flow taking into account the new ones added as part of the discussion phase which does change the numbering. When looking at the figures referenced in the reviews and the comments, please refer to the following updated figure numberings. Our new **Figure 19** has been moved into the main body of the paper as **Figure 5**, and hence **Figures 5-18** have all been shifted up by one to be **Figures 6-19**. The new **Table 6** has also been moved into the main body of the paper making it **Table 1**, hence making **Tables 1-5** into **Tables 2-6**. All other revisions to the paper are discussed in previous comments.
> > >
> > > - Figure 19 → Figure 5
> > >   - Figures 5-18 → 6-19
> > > - Table 6 → Table 1
> > >   - Tables 1-5 → 2-6

---

### Meta-Review · Area_Chair_w5mN · 2026-01-06

**Summary:**

This paper mainly presents that CroCo pretrained objectives learned implicit relighting, and this capability can be disentangled into lighting and intrinsic components, so it can be finetuned using small-scale in-domain data for applications. The major concerns from the reviewers are the lack of proper ablation study demonstrating the effectiveness of the pretrained weights, which have been made up in the edited version.  Reviewers also have the concerns on the paper scope and evaluation, and the AC believes the authors' rebuttal clarified most of the concerns and provided additional evidences to support the claim. Though having some limitations admitted by the authors, AC suggests to accept the paper as a poster presentation. More details are below.

**Reviewer Concerns:**

- Reviewer 3fbg provided strong supports on this paper. His concern on different encoders might be still outstanding, since the authors mainly emphasize the focus of the paper, but do not provide additional experiments to replace the encoder. However, given the strong claim that the cross-view completion benefits relighting learning, this one and other concerns from reviewer 3fbg are not blockers for acceptance.
- Reviewer FRcB have concerns that the paper is another version of relighting work, which has limited contributions to representation learning and not general enough. The authors clarified that the I / R / D are generally trained, and can be used for any downstream applications. So overall the method itself is learning a lighting-aware representation. AC believes the claim and feels the reviewers concern can be addressed. Another concern from reviewer FRcB is the data domain gap when using limited amount of synthetic data during training. The authors admit the difficulties of acquiring large amount of real data. But AC agrees with the authors that real data is hard to obtain, but can be still accumulated via constant collection, to mitigate the domain gap issues.  Overall, the remaining concerns from this reviewer do not block the acceptance.
- Reviewer Zxsi would like the paper to be an investigation-style one to answer a general question: how to learn photometric information from a pretraining objective. If the paper can be positioned in this way, AC also believes the impacts can be bumped up. The authors mainly argued that a probing paper can be developed based on the current paper, and the AC would love to see it in the future.
- Reviewer qx9H pointed out the issues of using a single global latent vector as a bottleneck while the authors did not fully address the concerns. The authors admitted the limitation but did not provide a stronger solution. The AC encourages the authors to highlight the limitations in the manuscript to reveal the difficulties. For shadow removal benchmark, the authors made a good point that the definition of shadow can be difference across benchmarks, while the purpose of this work is to showcase the generalization. AC agrees with that. The authors also did not fully address the nice-to-have requirements of architectural ablation like multiple lighting latents / spatial lighting maps. The concerns on the benefit from the pretrained encoders have been addressed by the additional ablation. Overall review qx9H mentioned many limitations of this work which are all admitted and clarified by the authors. AC believes those points are all very valuable, while additional experiments and more architecture exploration can be studied in future work.

**Reviewer Scores:**

- Reviewer 3fbg might not downgrade the score.
- Reviewer FRcB is possible to increase the score if the positioning of the paper is clear after reading the rebuttal.
- Reviewer Zxsi might not increase the score since there is a fundamental disagreement of the scope.
-  Reviewer qx9H might not increase the score if completeness is more important than solid contribution.

---

### Decision · Program_Chairs · 2026-01-26

Accept (Poster)